# Serum amyloid P component is an essential element of resistance against *Aspergillus fumigatus*

Andrea Doni[1], Raffaella Parente[1], Ilaria Laface[1,14], Elena Magrini[1], Cristina Cunha[2,3], Federico Simone Colombo[1], João F. Lacerda[4,5], António Campos Jr.[6], Sarah N. Mapelli[1], Francesca Petroni[1], Rémi Porte[1], Tilo Schorn[1], Antonio Inforzato[1,7], Toine Mercier[8,9], Katrien Lagrou[9,10], Johan Maertens[8,9], John D. Lambris[11], Barbara Bottazzi[1], Cecilia Garlanda[1,7], Marina Botto[12], Agostinho Carvalho[2,3] & Alberto Mantovani[1,7,13✉]

Serum amyloid P component (SAP, also known as Pentraxin 2; *APCS* gene) is a component of the humoral arm of innate immunity involved in resistance to bacterial infection and regulation of tissue remodeling. Here we investigate the role of SAP in antifungal resistance. $Apcs^{-/-}$ mice show enhanced susceptibility to *A. fumigatus* infection. Murine and human SAP bound conidia, activate the complement cascade and enhance phagocytosis by neutrophils. $Apcs^{-/-}$ mice are defective in vivo in terms of recruitment of neutrophils and phagocytosis in the lungs. Opsonic activity of SAP is dependent on the classical pathway of complement activation. In immunosuppressed mice, SAP administration protects hosts against *A. fumigatus* infection and death. In the context of a study of hematopoietic stem-cell transplantation, genetic variation in the human *APCS* gene is associated with susceptibility to invasive pulmonary aspergillosis. Thus, SAP is a fluid phase pattern recognition molecule essential for resistance against *A. fumigatus*.

[1] IRCCS Humanitas Research Hospital, via Manzoni 56, Rozzano Milan, Italy. [2] Life and Health Sciences Research Institute (ICVS), School of Medicine, University of Minho, Braga, Portugal. [3] ICVS/3B's-PT Government Associate Laboratory, Guimarães/Braga, Portugal. [4] Instituto de Medicina Molecular, Faculdade de Medicina de Lisboa, Lisboa, Portugal. [5] Serviço de Hematologia e Transplantação de Medula, Hospital de Santa Maria, Lisboa, Portugal. [6] Serviço de Transplantação de Medula Óssea (STMO), Instituto Português de Oncologia do Porto, Porto, Portugal. [7] Department of Biomedical Sciences, Humanitas University, Via Rita Levi Montalcini 4, Pieve Emanuele Milan, Italy. [8] Department of Hematology, University Hospitals Leuven, Leuven, Belgium. [9] Department of Microbiology, Immunology and Transplantation, KU Leuven, Leuven, Belgium. [10] Department of Laboratory Medicine and National Reference Centre for Mycosis, University Hospitals Leuven, Leuven, Belgium. [11] Department of Pathology and Laboratory Medicine, Perelman School of Medicine, University of Pennsylvania, Philadelphia, PA, USA. [12] Department of Immunology and Inflammation, Imperial College London, London, UK. [13] The William Harvey Research Institute, Queen Mary University of London, London, UK. [14]Present address: Department of Translational Medicine and for Romagna, University of Ferrara, Ferrara, Milan, Italy. ✉email: alberto.mantovani@humanitasresearch.it

The innate immune system represents the first line of resistance against pathogens and a key determinant in the activation and orientation of adaptive immunity through the complementary activities of a cellular and humoral arm[1]. Cell-associated innate immune molecules sense pathogen-derived agonists leading to activation of different inflammatory pathways[2]. Humoral pattern recognition molecules (PRMs) are essential components of the innate immune response sharing functional outputs with antibodies[3], including opsonisation, regulation of complement activation, agglutination, and neutralization[1]. Humoral PRMs in turn interact with and regulate cellular effectors[1,3–6]. Components of humoral innate immunity include complement components, ficolins, collectins, and pentraxins[1,7,8].

Pentraxins are an ancient group of proteins evolutionarily conserved from arachnids and insects to humans characterized by the presence of a 200 amino acid (aa) pentraxin domain and a pentraxin signature (HxCxS/TWxS, x = any aa)[7,8]. CRP (also called PTX1) and SAP (PTX2; *APCS* gene) are the classic short pentraxins. CRP and SAP are acute phase response proteins produced in the liver in response to infection and inflammatory cytokines[8–10].

SAP has complex biological functions that include regulation of matrix formation and resistance to infectious agents[8,10]. These general functions are shared by other pentraxins[11–14]. SAP was found to bind viruses (e.g., influenza)[15], bacteria (e.g., *Streptococcus pneumoniae*, *Staphylococcus aureus*, mycobacteria)[16,17] and malaria parasites[18]. Pentraxins, including SAP, behave as "ante-antibodies", activating complement and interacting with Fcγ receptors (Rs)[1–3]. SAP regulates complement activation by interacting with C1q[19], members of the ficolin family[20] and the complement regulator C4-binding protein (C4BP)[21]. Moreover SAP interacts with FcγRs and has opsonic activity[3,22–25]. In in vitro and in vivo models of infection, SAP has been reported to have dual influence on antimicrobial resistance depending on the species and experimental conditions. For instance, interaction of SAP with *S. pyogenes*, *Neisseria meningitides* and some variants of *Escherichia coli* led to decreased phagocytosis and killing by macrophages and inhibition of complement activation[26]. *Apcs*[−/−] mice showed better survival in experimental infections with *S. pyogenes* and *E. coli*[26].

SAP plays an important role in the regulation of extracellular matrix deposition. SAP is an important component of amyloidosis, where it promotes persistence of amyloid fibrils through binding and consequent stabilization[27,28]. A specific antibody against SAP promotes clearance of amyloid deposits in patients with systemic amyloidosis[29]. Moreover, SAP has been shown to inhibit lung fibrosis in preclinical models by regulating macrophage polarization[30–33].

SAP has entered early clinical assessment for the treatment of lung fibrosis[10]. In a randomized clinical trial, use of recombinant and identical to the endogenous human SAP (PRM-151) was associated with reduction in fibrocytes in pulmonary fibrosis patients[34]. A Phase 2 study trial of PRM-151, a novel anti-fibrotic immunomodulator, was reported in patients with Idiopathic Pulmonary Fibrosis (IPF)[35] and in a 28-week Phase 2 trial, infusions of SAP improved lung function[36,37] by inhibiting alternative activation of macrophages and fibrocyte differentiation[31,32].

The distant relative of SAP, PTX3, has been shown to play an essential role in resistance against *A. fumigatus* and related fungi. Strong genetic evidence is also consistent with a key role of PTX3 in resistance against *A. fumigatus* in humans in different clinical settings[38–47]. Given the PTX3 precedent, it was important to assess the role of SAP in resistance against *A. fumigatus*. Here we report that SAP was non-redundant for resistance against *A.*

*fumigatus* in mice. SAP-promoted phagocytosis of conidia involved activation of the complement cascade. In immunosuppressed mice, SAP administration resulted in protection against *A. fumigatus*. In the context of hematopoietic stem-cell transplantation, the *APCS* locus emerged as a novel risk factor regulating susceptibility to invasive pulmonary aspergillosis (IPA). Thus, SAP is a fluid phase PRM that plays an essential role in resistance against *A. fumigatus* infection and has therapeutic potential against fungal infections.

## Results

**SAP is essential for resistance to *A. fumigatus* infection.** *Apcs*[−/−] mice were used to assess the role of this molecule in resistance against *A. fumigatus* pulmonary infection. As shown in Fig. 1a, b, where two doses of conidia ($1 \times 10^8$ or $5 \times 10^7$) were injected intratracheally (i.t.), *Apcs*[−/−] mice showed increased susceptibility to infection. *Apcs*[−/−] mice showed lethal infection with a median survival time (MST) of 3 days compared to MST > 10 days of wt, both when $1 \times 10^8$ (Fig. 1a) or $5 \times 10^7$ (Fig. 1b) conidia were used. Actually, 89.9% (8/9) (Fig. 1a) and 44.4% (5/9) (Fig. 1b) of *Apcs*[−/−] mice succumbed on day 3 compared to 23.8% (2/9) or 0% (0/6) of wt mice. At the end of the experiment (monitoring until day 10), 11.1% (1/9) and 33.3% (3/9) of *Apcs*[−/−] mice survived to infection compared to 55.6% (5/9) or 83.3% (5/6) of wt mice, at $1 \times 10^8$ (Fig. 1a) or $5 \times 10^7$ (Fig. 1b) conidia ($P < 0.05$). Figure 1 shows one experiment out of 2 ($1 \times 10^8$) or 5 ($5 \times 10^7$) performed. Supplementary Table 1 summarizes cumulative results obtained in experiments using $5 \times 10^7$ conidia. Similar results were obtained in pulmonary infections by different clinically relevant species of the *Trichocomaceae* family, such as *A. flavus* ($5 \times 10^7$) (Supplementary Fig. 1) and *A. terreus* ($1 \times 10^8$) (Supplementary Fig. 2). Although *A. terreus* showed low pathogenicity in mice even with the highest dose of injected conidia ($1 \times 10^8$), differences in survival (90.9%, 10/11 of survived wt mice at $1 \times 10^8$ conidia compared to 61.5%, 8/13 of *Apcs*[−/−] mice; $P = 0.09$) (Supplementary Fig. 2a) and fungal burden in the lung ($P = 0.002$) (Supplementary Fig. 2b) indicated augmented susceptibility associated with SAP-deficiency. No relevance of SAP was observed in a model of fungal dissemination using *Candida albicans* (i.v., $5 \times 10^6$) (wt, 50.0% vs. *Apcs*[−/−], 58.3% of survived mice) (Supplementary Fig. 3).

In an effort to assess whether the phenotype observed in *Apcs*[−/−] mice was indeed related to SAP deficiency[48], recombinant murine SAP was used. Pre-opsonisation of *A. fumigatus* conidia with recombinant murine SAP rescued the susceptibility of *Apcs*[−/−] mice to infection, without affecting the resistance of wt mice (Fig. 1c). Indeed, 69% (9/13) of SAP-treated *Apcs*[−/−] mice resisted to infection (MST > 10 days; $P < 0.0001$) compared to 0% (0/14) of *Apcs*[−/−] mice treated with vehicle (MST 3 days). Survival was similar in wt groups (wt: 80%, 8/10; MST > 10 days; SAP-treated wt: 70%, 7/10; MST > 10 days). Similar results were observed when murine SAP (50 μg/mouse) was injected intravenously (i.v.) (Fig. 1d).

SAP-deficiency was associated with defective recruitment of neutrophils in response to *A. fumigatus*, as assessed by cell counts (Supplementary Fig. 4a) ($P = 0.008$; 16 h) and levels of myeloperoxidase (MPO) (*Apcs*[−/−], 449 ± 28 vs. wt, 564 ± 16 ng/ml; $P = 0.008$; 16 h) in bronchoalveolar lavage fluids (BALFs) (Supplementary Fig. 4b), and lower levels of C5a (*Apcs*[−/−], 22.7 ± 1.7 vs. wt, 31.5 ± 1.8 ng/ml; $P = 0.004$; 16 h) (Supplementary Fig. 4b). Decreased C3d in lung lysates was also observed ($P = 0.03$) (Supplementary Fig. 4c). Murine SAP pre-opsonisation restored the levels of MPO and the number of neutrophils in BALFs of *Apcs*[−/−] mice (16 h) (Supplementary Fig. 5), hence suggesting that SAP-opsonized conidia trigger an inflammatory response in the lung essential for resistance to fungus.

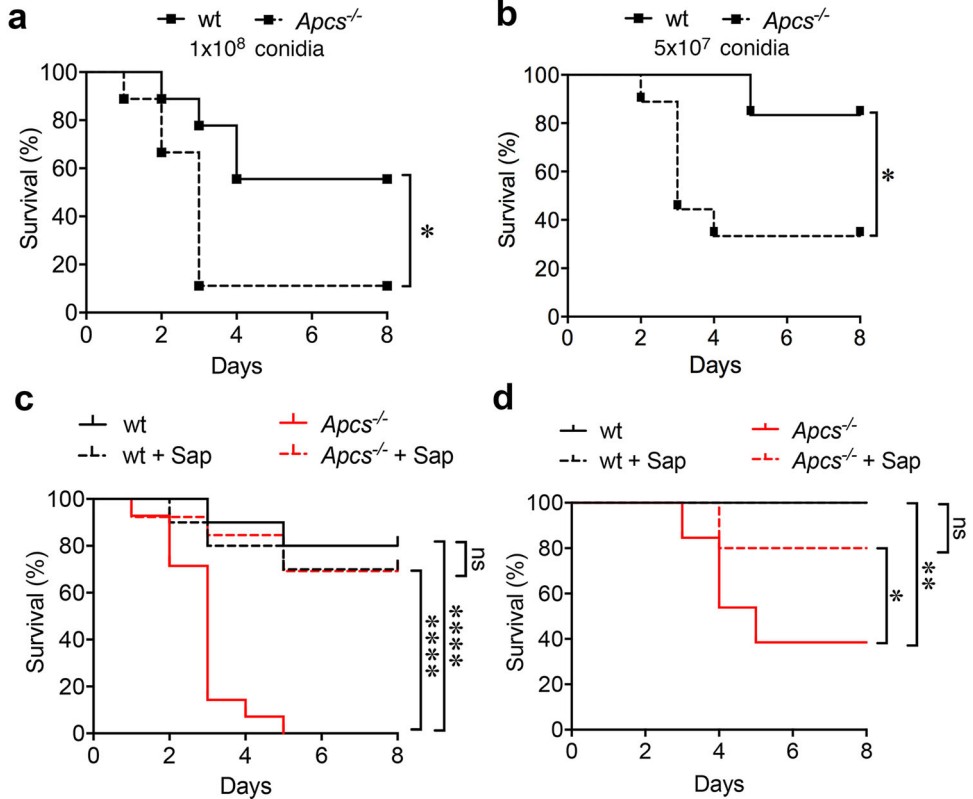

**Fig. 1 Susceptibility of *Apcs*⁻/⁻ mice to *A. fumigatus*. a, b** Percent survival of wt and *Apcs*⁻/⁻ mice after i.t. injection of $1 \times 10^8$ (**a**) or $5 \times 10^7$ (**b**) *A. fumigatus* (AF) conidia; wt, $n = 9$ (**a**) or n = 6 (**b**); *Apcs*⁻/⁻, $n = 9$ (**a, b**). **a** *$P = 0.02$; **b** $P = 0.05$ (Log-rank, Mantel-Cox test). **c, d** Rescue of susceptibility to *A. fumigatus* in *Apcs*⁻/⁻ mice. **c** Injection of non-opsonised or murine SAP(Sap)-opsonised AF conidia ($5 \times 10^7$). wt, $n = 10$; wt + Sap, $n = 10$; *Apcs*⁻/⁻, $n = 14$; *Apcs*⁻/⁻ + Sap, $n = 13$. ****$P < 0.0001$, wt vs. *Apcs*⁻/⁻ or *Apcs*⁻/⁻ vs. *Apcs*⁻/⁻ + Sap (Log-rank, Mantel-Cox test). **d** $5 \times 10^7$ and i.v. injection of Sap (50 µg/mouse). wt, $n = 10$; wt + Sap, $n = 10$; *Apcs*⁻/⁻, $n = 13$; *Apcs*⁻/⁻ + Sap, $n = 10$. **$P = 0.03$, wt vs. *Apcs*⁻/⁻; *$P = 0.05$, *Apcs*⁻/⁻ vs. *Apcs*⁻/⁻ + Sap (Log-rank, Mantel-Cox test).

**Mechanism of SAP-mediated resistance to *A. fumigatus*.** Injection of *A. fumigatus* conidia increased the circulating levels of SAP ($0.55 \pm 0.43$ µg/ml; $n = 3$) at 4 ($2.63 \pm 1.02$ µg/ml, $n = 4$) and 16 h ($22.12 \pm 8.50$ µg/ml, $n = 3$) comparably to those observed after LPS administration ($28.47 \pm 11.43$ µg/ml, $n = 3$; LPS, 0.8 mg/kg, 16 h) (Fig. 2a). In lungs ($n = 3$; 4 h), SAP localized in areas of cell recruitment and complement deposition closely associated with conidia (Fig. 2b).

Neutrophils are major players of resistance against *A. fumigatus*[49]. Therefore, we examined neutrophil phagocytosis of *A. fumigatus* in BALFs of *Apcs*⁻/⁻ mice. As shown in Fig. 2c, d, in the lungs of *Apcs*⁻/⁻ mice we observed a 6-fold increase of *A. fumigatus* CFU [*Apcs*⁻/⁻, median, $1.9 \times 10^8$, interquartile range (IQR) $3.0 \times 10^8 - 1.6 \times 10^8$ vs. wt, $4.0 \pm \text{x} \, 10^7$, IQR $6.8 \times 10^7 - 4.5 \times 10^7$; $P < 0.0001$] (Fig. 2c) and reduced phagocytosis by neutrophils (*Apcs*⁻/⁻, $28.74 \pm 3.96\%$ vs. wt, $42.53 \pm 6.39\%$; $P = 0.05$) (Fig. 2d).

Based on these in vivo results, the interaction of SAP with conidia and its impact on phagocytosis and complement activation was examined in vitro. Recombinant murine SAP bound viable dormant, swollen and germinated conidia, as assessed by FACS (Fig. 2e) and microscopy (Fig. 2f). A similar binding was observed for *A. flavus* and *A. terreus*, but not for *Candida albicans* (Supplementary Fig. 6a–c). Binding on *A. fumigatus* was competed by human SAP, but not by CRP (Fig. 2e). Similarly, PTX3 competed binding (Supplementary Fig. 6d), thus indicating that the binding site(s) on *A. fumigatus* is (are) conserved between mouse and human SAP and PTX3.

In whole blood, phagocytosis by neutrophils was reduced in *Apcs*⁻/⁻ mice, both at 1 (wt, $47.0 \pm 4.0$ vs. *Apcs*⁻/⁻, $41.3 \pm 2.4\%$;

$P = 0.05$) and 20 min (wt, $55.7 \pm 3.1$ vs. *Apcs*⁻/⁻, $44.4 \pm 2.3\%$; $P = 0.01$) after incubation with *A. fumigatus* conidia (Fig. 2g). Hence, SAP basal levels would be sufficient to affect conidia phagocytosis. Murine SAP pre-opsonisation rescued the defect (1 min, $P = 0.002$; 20 min, $P = 0.03$) and potentiated phagocytosis by wt neutrophils (1 min, $P < 0.0001$; 20 min, $P = 0.03$) (Fig. 2g). Similar results were obtained when human native SAP was used (Supplementary Fig. 7). In addition, membrane expression of CD11b in wt neutrophils was reduced from $97.2 \pm 0.4\%$ to $52.5 \pm 3.8\%$ and $47.4 \pm 2.8\%$, respectively, at 1 and 20 min after exposure to *A. fumigatus* (Fig. 2h). In neutrophils from *Apcs*⁻/⁻ mice, the decrease in CD11b expression was minor (basal, $97.0 \pm 0.5\%$; 1 min, $59.6 \pm 2.4\%$, $P = 0.07$; 20 min $57.7 \pm 2.2\%$, $P = 0.004$), as a consequence of an accumulation of CD11b in the phagocytic cup associated with phagocytosis[3,50,51]. Murine SAP pre-opsonisation restored CD11b internalization and increased it in wt (Fig. 2h). Therefore, SAP acts as an opsonin for early disposal of *A. fumigatus* by neutrophils, enhancing phagocytic activity.

The classical and lectin pathways are main initiators of complement activation against *A. fumigatus*[52]. Experiments of opsonophagocytosis were therefore performed in the presence of human or mouse sera deficient for different complement components to define the molecular mechanism responsible for SAP-mediated resistance to *A. fumigatus*. Phagocytosis by human neutrophils was abolished in serum depleted for C3 ($-72.0 \pm 4.7\%$, $P < 0.0001$), C1q ($-85.3 \pm 0.7\%$, $P < 0.0001$) and MBL ($-91.7 \pm 1.9\%$, $P < 0.0001$) compared to normal serum, thus indicating importance of both the classical and the MBL-mediated lectin pathway in resistance against this fungus. In

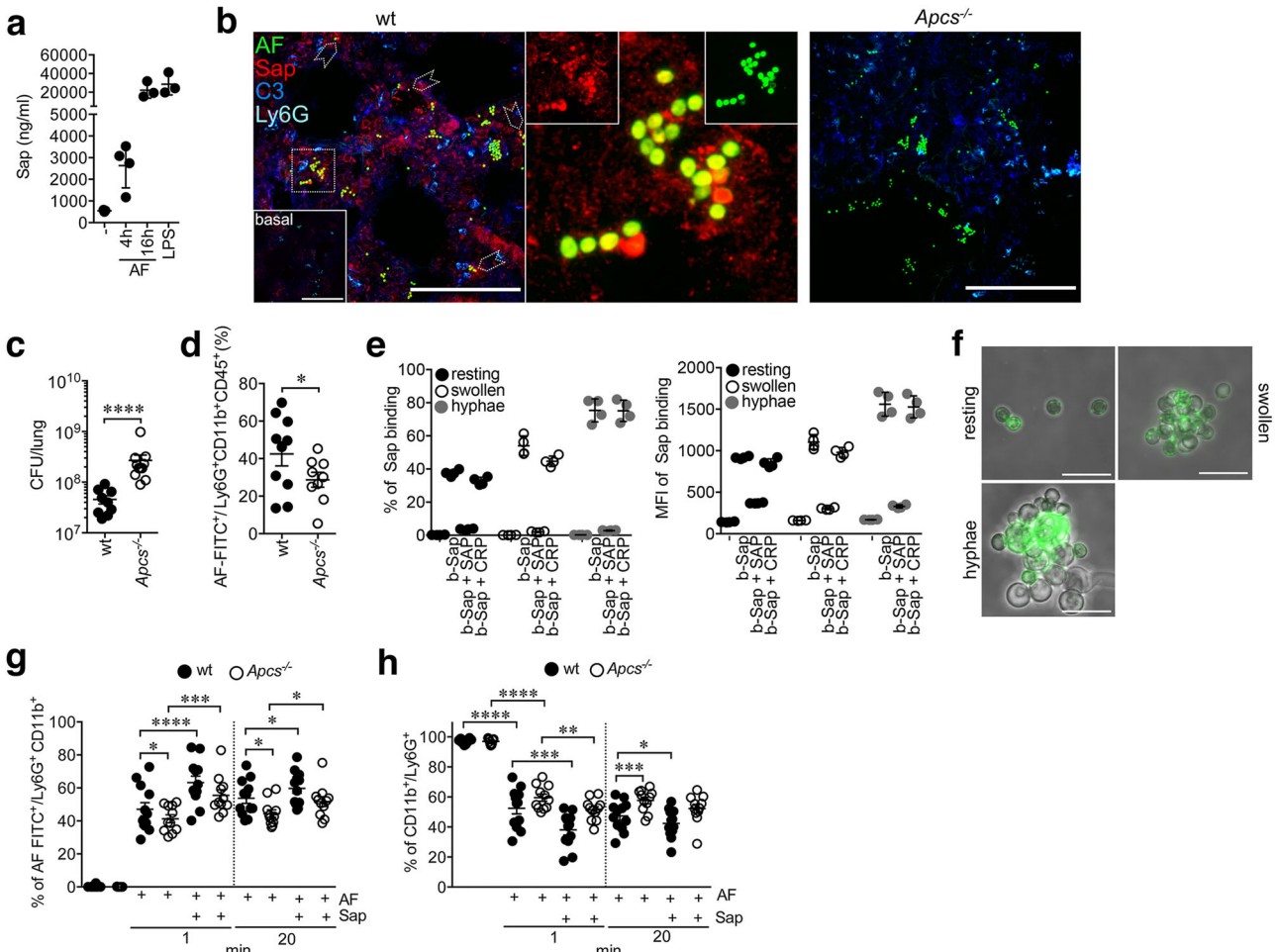

**Fig. 2 SAP acts as opsonin for phagocytosis of *A. fumigatus*. a** Induction of SAP levels in mice after i.t. injection of $5 \times 10^7$ AF conidia; −, sham mice; LPS, 0.8 mg/kg. Mean ± SD of an independent experiment; $n = 3$, −; $n = 4$, AF 4 h; $n = 3$, AF 16 h; $n = 3$, LPS. **b** Confocal images depicting SAP localization in infected lung (4 h; $5 \times 10^7$ fluorescein (FITC)-labeled AF conidia; $n = 3$ mice). Left, merged image of a lung stained for SAP (red), C3 (blue) and neutrophils (Ly6G+ cells; cyan). Conidia are evident from the detection of FITC (green) signal. Dashed white arrowheads indicate regions of SAP localization on AF conidia. A representative staining of lung of a non-infected mouse is also shown (basal; left box). Middle, close-up image of the white dashed region (left) showing SAP localization on AF. The image refers to extracted AF and SAP signals also separately shown in the boxes within the image. Right, a section of infected lung of *Apcs−/−* mouse is also shown. Scale bar, 50 μm. **c** Number of CFU per lung at 16 h after infection with $5 \times 10^7$ conidia. Each spot corresponds to a single mouse (wt, $n = 9$; *Apcs−/−* = 11). One experiment out of 2 performed with similar results. Mean ± SEM. $P < 0.0001$ (two-sided, Mann–Whitney *U*-test). **d** FACS analysis of in vivo phagocytosis in BALF neutrophils 4 h after injection of $5 \times 10^7$ FITC-labeled AF conidia. The figure shows results of two pooled experiments with $n = 10$ wt and $n = 9$ *Apcs−/−* mice. Mean ± SEM. *$P = 0.05$ (two-sided, unpaired *t*-test). **e, f** Interaction of SAP with *A. fumigatus*. **e** FACS analysis of binding of biotin-conjugated (b-) murine SAP (Sap; 10 μg/ml) to viable dormant or germinating conidia of AF ($1 \times 10^8$). Human SAP (50 μg/ml) and CRP (50 μg/ml) were also used. Mean ± SD of one quadruplicate experiment of two performed. **f** b-Sap (10 μg/ml) interaction with AF. Images refer to maximum intensity projection of *z*-stacks. Scale bar, 5 μm. **g, h** Effect of SAP on *A. fumigatus* phagocytosis by neutrophils. **g, h** FACS analysis (one experiment shown out of 3 performed) of phagocytosis (**g**) and CD11b internalization (**h**) in neutrophils after challenge with FITC-labeled AF conidia ($1 \times 10^7/200$ μl of blood) opsonized or not with murine SAP is shown. **g** neutrophil phagocytosis in whole blood of wt and *Apcs−/−* mice. **h** CD11b expression in neutrophils of **g**. **g, h** $n = 12$ wt and $n = 11$ *Apcs−/−* mice. Mean ± SEM. **g** 1 min, *$P = 0.05$; ***$P = 0.002$; ****$P < 0.0001$. 20 min, *$P = 0.01$, wt vs. *Apcs−/−*; *$P = 0.03$ wt vs. wt + Sap or *Apcs−/−* vs. *Apcs−/−* + Sap. **h** 1 min, *$P = 0.01$; ***$P = 0.0004$; ****$P < 0.0001$; 20 min, *$P = 0.04$; ***$P = 0.004$. **g, h** (two-sided, unpaired *t*-test).

two independent experiments using 5 and 10% of sera, opsonisation with human native SAP potentiated phagocytosis in normal (serum 5%, % of increase, 37.4 ± 10.3%, $P = 0.009$; serum 10%, 18.3 ± 2.0%, $P = 0.01$) and MBL-depleted serum (5%, 55.5 ± 17.0%; 10%, 19.8 ± 6.1%, $P = 0.04$), but not in sera depleted for C3 and C1q (Fig. 3a). Similarly, as assessed in independent experiments, neutrophil phagocytosis was reduced in whole blood of *C3−/−* (−43.4 ± 10.8%, $P = 0.02$), *C1q−/−* (−47.1 ± 11.4%, $P = 0.02$) and *Mbl1/2−/−* (−40.2 ± 12.3%, $P = 0.01$) mice compared to wt, but not in *Fb−/−* mice (Fig. 3b). Pre-opsonization with murine SAP potentiated

phagocytosis in wt [% of increase, 30.8 ± 5.4% (Fig. 3b, left); 94.3 ± 48.2%, $P = 0.004$ (Fig. 3b, middle); 134.7 ± 45.2%, $P = 0.02$ (Fig. 3b, right)], *Mbl1/2−/−* (68.6 ± 28.3%, $P = 0.002$) and *Fb−/−* (196.6 ± 61.8%, $P = 0.005$)] but not in *C3−/−* and *C1q−/−* mice (Fig. 3b). These results indicate that the interaction with the classical complement pathway is required for the initiation of SAP-mediated phagocytosis, with no relevance of the alternative pathway. In agreement, a decreased C1q deposition on *A. fumigatus* conidia was observed after incubation with plasma from *Apcs−/−* mice ($P = 0.05$) (Fig. 3c). Moreover, while MBL in serum plays a role

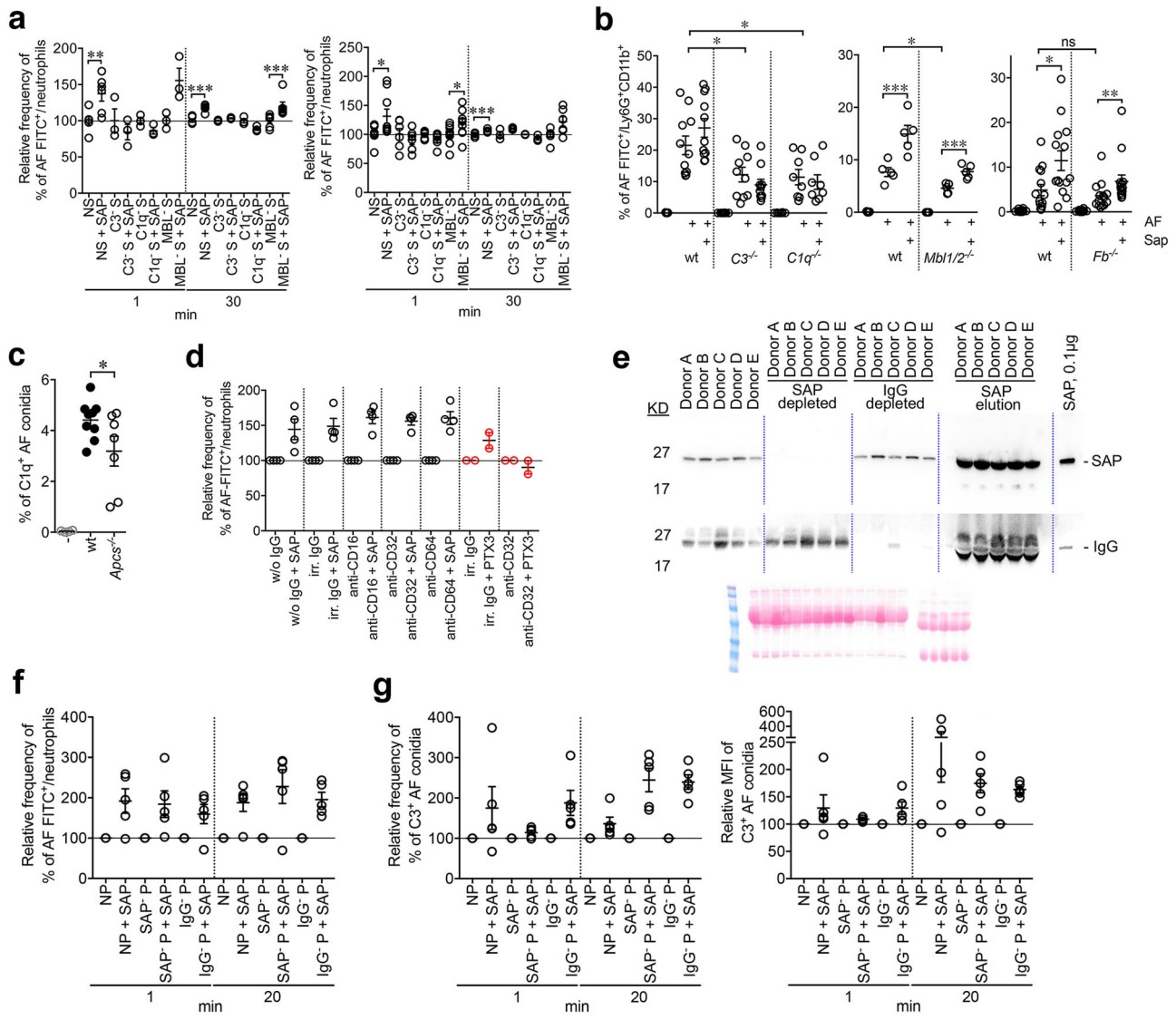

**Fig. 3 Initiation of classical complement activation at the bases of SAP-mediated phagocytosis. a** FITC-labeled AF conidia ($1.6 \times 10^6$) phagocytosis by freshly isolated human neutrophils ($2 \times 10^5$) in presence of sera depleted ($-$) of complement components and the effect of human SAP opsonisation after 1 and 30 min. AF conidia ($2 \times 10^8$) opsonisation with 100 μg of human native SAP. Five percent (left) or 10% (right) of human sera were used. Mean ± SEM of two experiments performed in triplicate and merged. Each spot corresponds to a replicate. Left, NS vs. NS + SAP, **$P = 0.009$; ***$P = 0.002$. MBL$^-$ S vs. MBL$^-$ S + SAP, ***$P = 0.002$ (two tailed, Mann–Whitney $U$-test). Right, NS vs. NS + SAP, *$P = 0.01$; ***$P = 0.002$. MBL$^-$ S vs. MBL$^-$ S + SAP, *$P = 0.04$ (two tailed, Mann–Whitney $U$-test). **b** FACS analysis of phagocytosis of FITC-labeled AF conidia ($5 \times 10^6$/200 μl of blood; 20 min) opsonized or not with murine SAP in whole blood of wt, C3-, C1q-, MBL1/2-, and Factor B-deficient mice. Each figure refers to independent experiments performed. Results of $C3^{-/-}$, $C1q^{-/-}$, and $Mbl1/2^{-/-}$ are mean ± SEM of one experiment, whereas for $Fb^{-/-}$ two experiments were merged. Correspondent wt littermates were used as control for C1q-, MBL1/2- (C57BL/6J) and Factor B-deficient mice (C57BL6/NJ). Each spot represents a single mouse. Left, $n = 11$ wt, $n = 10$ $C3^{-/-}$, $n = 8$ $C1q^{-/-}$. wt vs. $C3^{-/-}$, *$P = 0.02$; wt vs. $C1q^{-/-}$, *$P = 0.02$ (two-sided, unpaired $t$-test). Middle, $n = 5$ wt, $n = 5$ $Mbl1/2^{-/-}$. wt vs. $Mbl1/2^{-/-}$, *$P = 0.01$; wt vs. wt + Sap, ***$P = 0.004$; $Mbl1/2^{-/-}$ vs. $Mbl1/2^{-/-}$ + Sap, ***$P = 0.002$ (two-sided, unpaired $t$-test). Right, $n = 12$ wt, $n = 14$ $Fb^{-/-}$. wt vs. wt + Sap, *$P = 0.02$ (two-sided, unpaired $t$-test); $Fb^{-/-}$ vs. $Fb^{-/-}$ + Sap, **$P = 0.005$ (two tailed, Mann–Whitney $U$-test). **c** FACS analysis of C1q deposition on AF conidia ($1 \times 10^7$) in presence of plasma from wt ($n = 9$) or $Apcs^{-/-}$ ($n = 7$) mice. Mean ± SEM; *$P = 0.05$ (two-sided, unpaired $t$-test). **d** Effect of blocking of Fcγ receptors in SAP-mediated phagocytosis. Human SAP-pre-opsonised FITC-labeled AF conidia ($1.6 \times 10^6$) phagocytosis by freshly isolated human neutrophils ($2 \times 10^5$; 5 min) in presence of NS and anti-CD16, anti-CD32, anti-CD64 or an irrelevant mouse IgG1 (all used at 10 μg/ml). Each spot corresponds to a single healthy donor ($n = 4$) performed in duplicate. Mean ± SEM. **e–g** Effect of IgG depletion in SAP-mediated phagocytosis (**f**) and complement deposition (**g**). **e** Western blot analysis for SAP and IgGs in human plasma and after their depletion. 1 μl/lane of plasma-heparin from five healthy donors (A–E) and 10 μg/lane of proteins eluted from the anti-SAP column were loaded. Ponceau red staining was shown. Human SAP (0.1 μg/lane) was used as control. **f** FACS analysis of phagocytosis by neutrophils ($2 \times 10^5$) isolated from A–E donors after challenge with FITC-labeled AF conidia ($1.6 \times 10^6$) opsonized or not with human SAP (100 μg/$1 \times 10^9$ conidia/ml) in the presence of autologous plasma-heparin (10%) depleted of SAP (SAP$^-$) and IgGs (IgG$^-$). **g** C3 deposition on AF conidia ($1 \times 10^7$) opsonized or not with human SAP. SAP$^-$ and IgG-depleted plasma-heparin (10%) from A–E donors were used. **f**, **g** Each spot refers to mean of a duplicate per donor. Mean ± SEM.

in opsonisation, MBL and ficolins, the latter ascertained using specific antibodies (Supplementary Fig. 8), are dispensable for SAP-mediated enhanced phagocytosis.

It was important to assess whether the opsonic activity of SAP was dependent on FcγR or IgG. SAP-mediated opsonophagocytosis did not involve engagement of FcγRs, as ascertained in phagocytosis experiments with specific blocking antibodies (Fig. 3d). As control[51], PTX3-mediated phagocytosis was inhibited by FcγRII blockage (Fig. 3d). Depletion of IgGs from the plasma of healthy donors ($n = 5$) (Fig. 3e) resulted in decreased neutrophil phagocytosis at the 20 min time-point ($-26.6 \pm 3.3\%$, $P = 0.0006$) compared to control plasma. A similar reduction was observed when plasma depleted from SAP (Fig. 3e) was used (1 min, $-1.65 \pm 13.7\%$, $P = 0.28$; 20 min, $-20.5 \pm 5.9\%$, $P = 0.04$). Human SAP pre-opsonization rescued and potentiated neutrophil phagocytosis in IgG-depleted (% of increase, 1 min, $59.6 \pm 19.5\%$, $P = 0.04$; 20 min, $95.6 \pm 14.6\%$, $P = 0.0007$), SAP-depleted (1 min, $84.1 \pm 26.9\%$, $P = 0.008$; 20 min, $127.1 \pm 34.1\%$, $P = 0.02$) and normal (1 min, $92.0 \pm 24.7\%$; 20 min, $87.9 \pm 17.9\%$, $P = 0.008$) plasma (Fig. 3f), thus indicating that SAP-mediated opsonophagocytosis did not involve an antibody-mediated engagement of FcγR.

SAP induced complement activation on *A. fumigatus* culminating in the formation of the membrane attachment complex (MAC; C5b-C9). In plasma of *Apcs*$^{-/-}$ mice, decreased C3 deposition was observed on *A. fumigatus* conidia in vitro (Fig. 4a).

Pre-opsonisation of *A. fumigatus* with recombinant murine SAP rescued and increased C3 deposition in *Apcs*$^{-/-}$ and wt plasma, respectively (Fig. 4a). In the same experiment, a decreased C5a formation was observed in plasma of *Apcs*$^{-/-}$ compared to wt mice (1 min, $8.5 \pm 2.1$ vs. $13.0 \pm 4.5$ ng/ml; 5 min, $14.2 \pm 4.3$ vs. $35.7 \pm 6.9$ ng/ml, $P = 0.008$; 10 min, $26.5 \pm 2.2$ vs. $53.2 \pm 7.5$ ng/ml, $P = 0.003$; 20 min, $61.5 \pm 7.8$ vs. $113.8 \pm 15.5$ ng/ml, $P = 0.005$, respectively, *Apcs*$^{-/-}$ vs. wt) (Fig. 4b). This defect was rescued by murine SAP opsonisation (1 min, $P = 0.003$; 5 min, $P = 0.007$; 10 min, $P = 0.002$; 20 min, $P = 0.04$), which also increased C5a in wt plasma (Fig. 4b). Assembly of MAC on the surface of *A. fumigatus* conidia was decreased in sera depleted for C3 (1 min, $-72.2 \pm 4.8\%$; 30 min, $-73.0 \pm 10.1\%$; Mean ± SD; $P < 0.0001$), C1q (1 min, $-81.8 \pm 3.6\%$; 30 min, $-68.5 \pm 4.0\%$; $P < 0.0001$), MBL (1 min, $-92.2 \pm 2.8\%$, $P < 0.001$; 30 min, $-84.5 \pm 3.0\%$, $P < 0.0001$) and FB (1 min, $-80.6 \pm 5.2\%$; 30 min, $-46.4 \pm 1.0\%$; $P < 0.0001$) and increased in FH-depleted serum (1 min, $+5.6 \pm 15.8\%$; 30 min, $+55.0 \pm 24.9\%$; $P < 0.005$). Pre-opsonisation with human SAP enhanced MAC formation on conidia both at 1 min ($P = 0.001$) and at 30 min ($P = 0.008$) in the presence of normal serum and in those depleted of MBL, FB, and FH, but not in serum depleted for C1q (Fig. 4c). As assessed in a resazurin-based cell viability assay, pre-opsonisation of human SAP resulted in increased microbicidal activity of *A. fumigatus* conidia in normal serum and in serum depleted of MBL (Fig. 4d). The effect was abolished in human sera depleted of C3 or C1q. Posaconazole

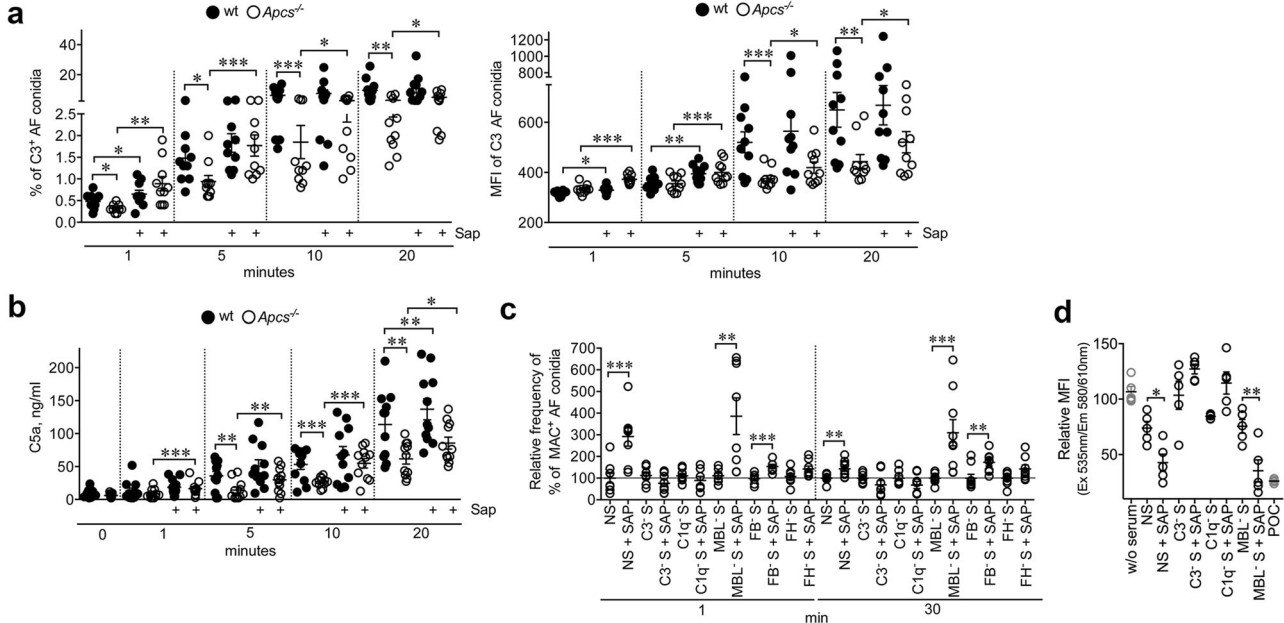

**Fig. 4 SAP-mediated complement activation on *A. fumigatus*. a** FACS analysis (frequency, left; MFI, right) of C3 from wt (−, $n = 10$; + Sap, $n = 10$) and *Apcs*$^{-/-}$ (−, $n = 10$; + Sap, $n = 10$) mouse plasma on AF conidia ($1 \times 10^7$) opsonized or not with murine SAP. **a** Left, wt vs. *Apcs*$^{-/-}$, *$P = 0.01$ (1 min), *$P = 0.02$ (5 min), ***$P = 0.003$ (10 min), **$P = 0.007$ (20 min). wt vs. wt + Sap, *$P = 0.02$ (1 min). *Apcs*$^{-/-}$ vs. *Apcs*$^{-/-}$ + Sap, **$P = 0.008$ (1 min), ***$P = 0.003$ (5 min), *$P = 0.05$ (10 min), *$P = 0.05$ (20 min) (two-sided, unpaired *t*-test). **a** Right, wt vs. *Apcs*$^{-/-}$, ***, $P = 0.002$ (10 min), **$P = 0.006$ (20 min). wt vs. wt + Sap, *$P = 0.01$ (1 min), **$P = 0.005$ (5 min). *Apcs*$^{-/-}$ vs. *Apcs*$^{-/-}$ + Sap, ***$P = 0.0003$ (1 min), ***$P = 0.002$ (5 min), *$P = 0.04$ (10 min and 20 min) (two-sided, unpaired *t*-test). **b** Murine plasma C5a levels after incubation with AF conidia shown in **a**. $n = 11$ mice/genotype. wt vs. *Apcs*$^{-/-}$, **$P = 0.008$ (5 min), ***$P = 0.003$ (10 min), **$P = 0.005$ (20 min). wt vs. wt + Sap, **$P = 0.009$ (20 min). *Apcs*$^{-/-}$ vs. *Apcs*$^{-/-}$ + Sap, ***$P = 0.003$ (1 min), **$P = 0.007$ (5 min), ***$P = 0.002$ (10 min), *$P = 0.04$ (20 min) (two-sided, unpaired *t*-test). **a**, **b** A representative independent experiment shown out of 3 performed. Each spot represents plasma from a single mouse. **a**, **b** Mean ± SEM. **c** FACS analysis of SAP-mediated MAC deposition on AF conidia ($1 \times 10^7$) in the presence of human sera depleted of complement components. Mean ± SEM of two experiments in quadruplicate and merged out of 4 performed. Each spot corresponds to a replicate. NS vs. NS + SAP, ***$P = 0.001$; **$P = 0.008$ (two-sided, unpaired *t*-test). MBL⁻ S vs. MBL⁻ S + SAP, **$P = 0.006$; ***$P = 0.003$ (two-sided, unpaired *t*-test). FB⁻ S vs. FB⁻ S + SAP, ***$P = 0.002$; **$P = 0.009$ (two-sided, unpaired *t*-test). **d** Quantitative analysis of AF viability performed using a resazurin-based assay. AF viability assay performed in sera depleted of complement components. Results are mean ± SEM of the relative fluorescence intensity. Effect of Posaconazole (POC; 1 µM) exposure is also shown. Figure summarizes one experiment out of 5 performed with $n = 5$ replicates each. *$P = 0.01$; **$P = 0.009$ (two-sided, unpaired *t*-test).

(POC) was used as antifungal control (Fig. 4d). Actual fungal killing was also ascertained as CFU counts (Supplementary Fig. 9). SAP-mediated complement activation on *A. fumigatus* was antibody-independent. Decreased C3 deposition was observed in IgG-depleted plasma, both at 1 ($-53.5 \pm 7.9\%$, $P = 0.02$) and 20 min ($-74.2 \pm 4.7\%$, $P = 0.002$) compared to control. In SAP-depleted plasma, reduction of C3 deposition corresponded to $-30.5 \pm 9.1\%$, (1 min, $P = 0.025$) and $-58.6 \pm 4.9\%$, (20 min, $P < 0.0001$). Pre-opsonisation with human SAP rescued and increased C3 deposition in both IgG- (% of increase, 1 min, $87.9 \pm 25.2\%$, $P = 0.022$; 20 min, $139.6 \pm 14.6\%$, $P < 0.0001$) and SAP-depleted (1 min, $14.3 \pm 4.4\%$; 20 min, $144.5 \pm 23.8\%$, $P = 0.001$) plasma as well as in the control (1 min, $75.0 \pm 43.4\%$; 20 min, $36.1 \pm 13.3\%$, $P = 0.008$) (Fig. 3g), thus suggesting an effect of SAP in promoting complement activation on *A. fumigatus* under mimed immunosuppressive conditions of antibody deficiencies.

Having found that SAP plays a non-redundant role in resistance to *A. fumigatus* and that neutrophils were involved, it was important to explore the role and significance of other phagocytic cell populations using in vitro, ex vivo and in vivo approaches. As shown in Fig. 5a, in a whole blood assay, in vitro SAP increased the phagocytic activity of neutrophils [% of increase, wt, $134.3 \pm 19.6\%$, $P < 0.0001$ (1 min); $103.8 \pm 25.4\%$, $P = 0.003$ (20 min); *Apcs*$^{-/-}$, $176.3 \pm 34.2\%$, $P = 0.0004$ (1 min); $110.3 \pm 311.0\%$, $P < 0.0001$ (20 min)] and monocytes [wt, $138.2 \pm 12.7\%$, $P < 0.0001$ (1 min); $83.1 \pm 14.5\%$, $P = 0.002$ (20 min); *Apcs*$^{-/-}$, $104.4 \pm 20.9\%$, $P = 0.005$ (1 min); $148.3 \pm 23.7\%$, $P < 0.0001$ (20 min)], but not of DC (Fig. 5a). Similar results were obtained with human myelomonocytic cells in vitro (30 min; $P = 0.03$) (Fig. 5b).

Bone marrow-derived myelomonocytic cells were used to further investigate the impact of SAP on this compartment. In the presence of correspondent serum (10%), bone marrow-derived macrophages obtained from *Apcs*$^{-/-}$ mice produced decreased levels of CXCL1 (*Apcs*$^{-/-}$, $7.5 \pm 2.0$ vs. wt, $8.6 \pm 1.1$ ng/ml), CXCL2 (*Apcs*$^{-/-}$, $15.6 \pm 7.2$ vs. wt, $29.8 \pm 3.7$ ng/ml), TNF-α (*Apcs*$^{-/-}$, $3.7 \pm 0.6$ vs. wt, $4.5 \pm 0.4$ ng/ml), IL-6 (*Apcs*$^{-/-}$, $3.1 \pm 0.8$ vs. wt, $4.7 \pm 0.9$ ng/ml) after stimulation with *A. fumigatus* conidia (24 h) (Fig. 5c).

We next investigated the impact of the *Apcs* deficiency on macrophage and DC function in vivo. As assessed by FACS, injection of conidia pre-opsonized with murine SAP increased phagocytosis (4 h) in BALF macrophages (% of increase, wt, $41.8 \pm 3.4\%$, $P = 0.004$; *Apcs*$^{-/-}$, $26.8 \pm 5.0\%$, $P = 0.03$) and neutrophils (wt, $81.5 \pm 32.6\%$, $P = 0.03$; *Apcs*$^{-/-}$, $285.9 \pm 60.2\%$, $P = 0.0003$) derived from wt and *Apcs*$^{-/-}$ mice, but not in DCs (Fig. 5d). SAP-deficiency did not affect phagocytosis in BALF macrophages ($51.4 \pm 4.9$ vs. $51.0 \pm 14.1\%$; *Apcs*$^{-/}$ vs. wt) and DCs ($7.8 \pm 1.5$ vs. $11.2 \pm 3.4\%$; $P = 0.18$), but reduced it in neutrophils ($8.1 \pm 2.5$ vs. $16.0 \pm 3.7\%$, *Apcs*$^{-/}$ vs. wt, % of decrease: $-49.2 \pm 15.5\%$; $P = 0.05$) (Fig. 5d). In conclusion, although SAP facilitates recognition, phagocytosis and functional responses by macrophages, the predominant effector cells affected by *Apcs* deficiency in vivo are neutrophils.

Collectively, these results indicate that SAP-mediated resistance against *A. fumigatus* involves complement-mediated opsonization, neutrophil phagocytosis and pathogen killing.

**Therapeutic potential of SAP.** Having established that SAP is an essential component of resistance to *A. fumigatus*, it was important to obtain indications as to its therapeutic potential. *A. fumigatus* infection is a clinical challenge in patients with congenital or acquired immune deficiency. The most important risk factor for IPA is represented by neutropenia and monocytopenia that occur in immune-compromised patients[40]. The potential therapeutic effect of SAP was determined in transiently myelosuppressed mice. Therefore, we used a model of cyclophosphamide-induced immune suppression to obtain indications as to the therapeutic potential of SAP. Dosage of *A. fumigatus* conidia was newly optimized in mice after 2-day treatment with cyclophosphamide (150 mg/kg). Human native SAP (4 mg/kg) was injected intraperitoneally (i.p.) at 2 and 24 h after infection. Immune-compromised mice did not survive infection with $5 \times 10^7$ (17/17; MST 4 days), $1 \times 10^7$ (7/7; MST 4 days) and $5 \times 10^6$ (8/8; MST 5 days) conidia (Fig. 6a). SAP protected immune-compromised mice increasing survival to 20% ($5 \times 10^7$; 12/15, MST 4; $P = 0.002$), 62.5% ($1 \times 10^7$; 3/8, MST > 10; $P = 0.001$) and 80% ($5 \times 10^6$; 2/10, MST > 10; $P = 0.004$), respectively (Fig. 6a). A 12-fold reduction in fungal burden was observed in the lungs of SAP-treated mice 16 h after infection with $1 \times 10^7$ conidia (median $2.0 \times 10^6$ CFU, IQR $9.0 \times 10^6 - 8.0 \times 10^6$ vs. $4.4 \times 10^7$ CFU, IQR $9.1 \times 10^7 - 7.5 \times 10^7$; $P = 0.02$) (Fig. 6b). Similarly, SAP (50 μg/mouse) administered i.t. 2 h after infection ($5 \times 10^6$ conidia) protected immune-compromised mice. 2/13 untreated mice survived (15.4%; MST 7 days), whereas 9/12 SAP-treated mice survived (75.0%; MST > 10; $P = 0.003$) (Fig. 6c). I.t. SAP treatment resulted in a 23-fold reduction of lung CFU 16 h after infection (median $2.3 \times 10^4$, IQR $1.1 \times 10^5 - 1.0 \times 10^5$ vs. $5.3 \times 10^6$, IQR $9.5 \times 10^6 - 7.7 \times 10^6$; $P = 0.008$) (Fig. 6d).

**Genetic variation in human SAP influences the risk of IPA.** In view of the critical role of SAP during fungal infection in vivo, we next investigated the relationship between genetic variation in the human *APCS* gene and susceptibility to IPA in patients at-risk. For such, we assessed the cumulative incidence of IPA in patients from the IFIGEN cohort undergoing allogeneic hematopoietic stem-cell transplantation (Supplementary Table 2) according to recipient or donor *APCS* genotypes of the single nucleotide polymorphisms (SNPs) rs2808661 and rs3753869 (Table 1). These SNPs were selected based on their ability to tag surrounding variants with a pairwise correlation coefficient $r^2$ of at least 0.80 and a minor allele frequency ≥5%, covering therefore the whole genetic variability across the *APCS* locus (Supplementary Fig. 10).

The rs2808661 SNP in the donor genome was associated with an increased risk of IPA after transplantation, whereas recipient rs2808661 displayed a trend towards increased risk (Fig. 7a). The cumulative incidence of IPA for donor rs2808661 was 23% for GG, 21% for GA ($P = 0.67$) and 56% for AA genotypes ($P = 0.04$), whereas for recipient rs2808661, the cumulative incidence ranged from 20% for GG to 25% for GA ($P = 0.58$) and 36% for AA ($P = 0.08$) (Fig. 7a). Upon modeling a dominant mode of inheritance, the key contribution of the donor AA genotype to the risk of infection was confirmed. As for the rs3753869 SNP, association with risk was only detected in donor genomes (Fig. 7b). The cumulative incidence of IPA was 21% for CC, 26% for CA ($P = 0.30$) and 43% for AA genotypes ($P = 0.03$), respectively.

In a multivariate model accounting for age, gender, type of transplantation, acute graft-versus-host disease (GVHD) and antifungal prophylaxis, the AA genotype in rs3753869 conferred a 3.6-fold increased risk of developing IPA (Table 2). Collectively, these results highlight genetic variation at the *APCS* locus as a novel risk factor regulating susceptibility to IPA.

To assess whether genetic variation at the *APCS* locus was associated with different SAP levels, this pentraxin was measured in samples from hematological patients from the FUNBIOMICS cohort[53]. SAP serum levels were similar in IPA patients ($18.6 \pm 1.2$ μg/ml, $n = 20$) compared to controls ($16.6 \pm 1.3$ μg/ml, $n = 14$; $P = 0.2$), whereas higher amounts were present in BALFs ($65.7 \pm 26.4$ ng/ml, $n = 9$; vs. $2.6 \pm 0.7$ ng/ml, $n = 7$; $P = 0.0007$) (Fig. 7c).

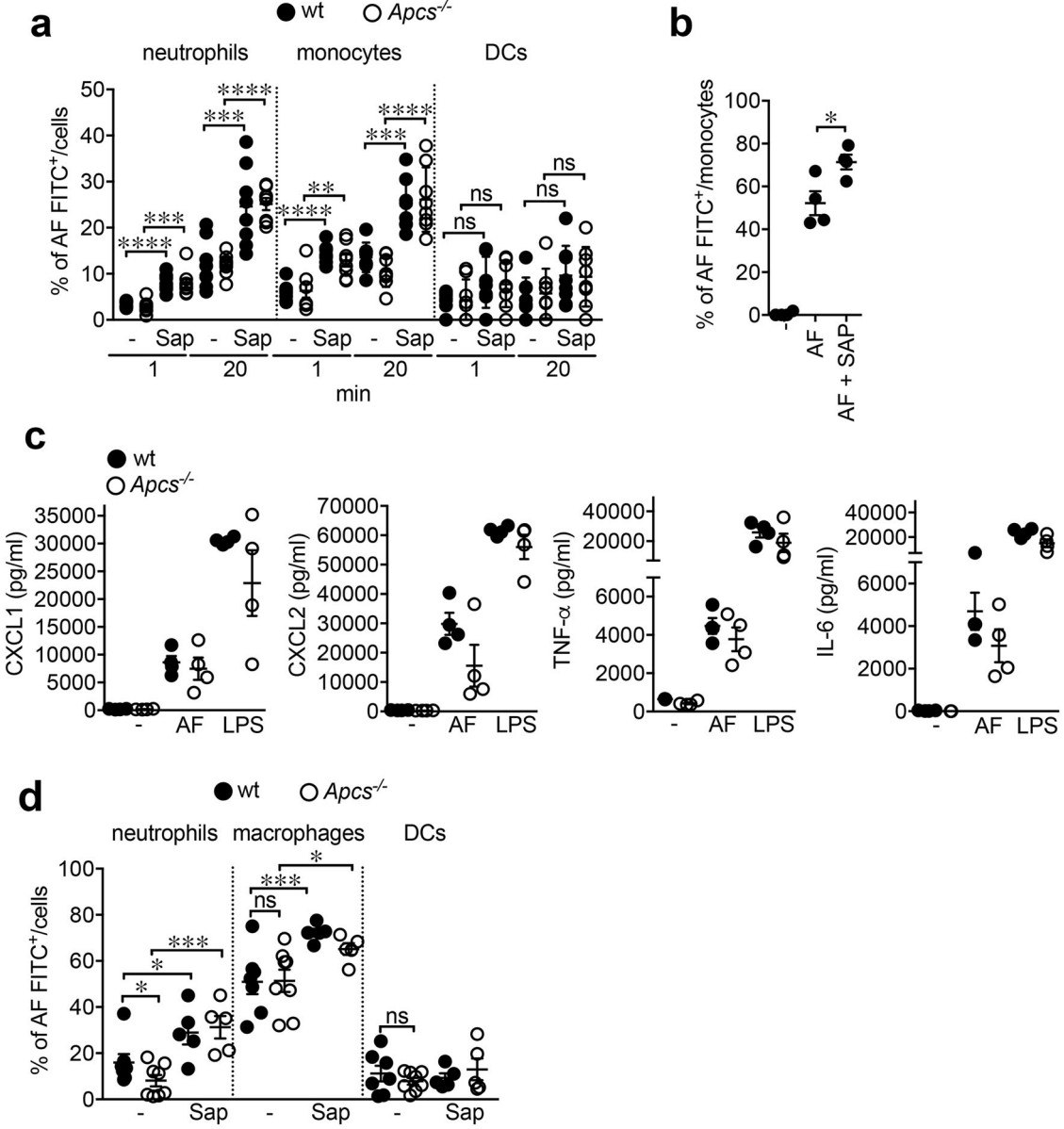

**Fig. 5 Effects of SAP on myelomonocytic cell function. a** FACS analysis of phagocytosis (1 and 20 min) of FITC-labeled AF conidia opsonized with murine SAP by peripheral murine neutrophils (Ly6G[+] CD11b[+]), monocytes (Ly6C[+] CD11b[+]) and DCs (CD11c[+]) (1×10[7] conidia/200 µl of blood). N = 7 wt, n = 8 Apcs[−/−]. wt vs. wt + Sap, ****P < 0.0001 (1 min), ***P = 0.003 (20 min) (neutrophils); ****P < 0.0001 (1 min), ***P = 0.002 (20 min) (monocytes). Apcs[−/−] vs. Apcs[−/−] + Sap, ***P = 0.0004 (1 min), ****P < 0.0001 (20 min) (neutrophils); **P = 0.005 (1 min), ****P < 0.0001 (20 min) (monocytes) (two-sided, unpaired t-test). **b** Effect of human SAP opsonisation on in vitro phagocytosis by isolated human monocytes (n = 4 donors) (1×10[5] cells and 1×10[6] FITC-labeled AF conidia; 30 min; 10% of autologous serum). *P = 0.03 (two-sided, unpaired t-test). **c** Levels of CXCL1, CXCL2, TNF-α, and IL-6 in supernatants of bone marrow-derived macrophages from wt and Apcs[−/−] mice stimulated (24 h) with A. fumigatus conidia (ratio 1:5, cell:conidia) in the presence of correspondent mouse serum (10%). LPS, 10 ng/ml. n = 4 mice per genotype. **d** FACS analysis of FITC-labeled AF conidia phagocytosis by neutrophils (CD45[+] Ly6G[+]), macrophages (CD45[+] Ly6C[+] F4/80[+]) and DCs (CD45[+] CD11c[+]) collected from BALFs of wt (−, n = 7; Sap, n = 5) and Apcs[−/−] (−, n = 8; Sap, n = 5) mice after i.t. injection of 5 × 10[7] conidia (4 h) opsonized with murine SAP. **d** Neutrophils, wt vs. Apcs[−/−], *, P = 0.05; wt vs. wt + Sap, *, P = 0.03; ***, P = 0.0003. Macrophages, *P = 0.030; ***P = 0.004 (two-sided, unpaired t-test). **a–d** Each spot corresponds to a single mouse (**a, c, d**) or single healthy donor (**b**). **a–d** Mean ± SEM of independent experiments.

Once analyzed for the single donor genotype, SAP concentration in sera was to 18.8 ± 1.7 µg/ml for GG (n = 12), 23.7 ± 0.5 µg/ml for GA (n = 3) and 14.7 ± 0.9 µg/ml for AA (n = 4; P = 0.103, AA vs. GG; P = 0.0006, AA vs. GA; P = 0.05, AA vs. GG+GA) in donor rs2808661 (Fig. 7d). Regarding donor rs3753869, SAP concentration was 19.1 ± 1.3 µg/ml for CC (n = 13), 23.17 ± 2.7 µg/ml for CA (n = 3) and 13.3 ± 2.9 µg/ml for AA (n = 4; P = 0.03, AA vs. CC; P = 0.06, AA vs. CA; P = 0.02, AA vs. CC+CA) (Fig. 7e). No differences were observed for GG vs. GA and

CC vs. CA, respectively, in recipient rs2808661 and rs3753869 (Supplementary Fig. 11). Therefore, the increased risk of IPA according to donor APCS rs2808661 and rs3753869 genotypes is associated with lower serum levels of SAP.

## Discussion

The results presented here show that SAP is an essential component of resistance against the fungal pathogen A. fumigatus.

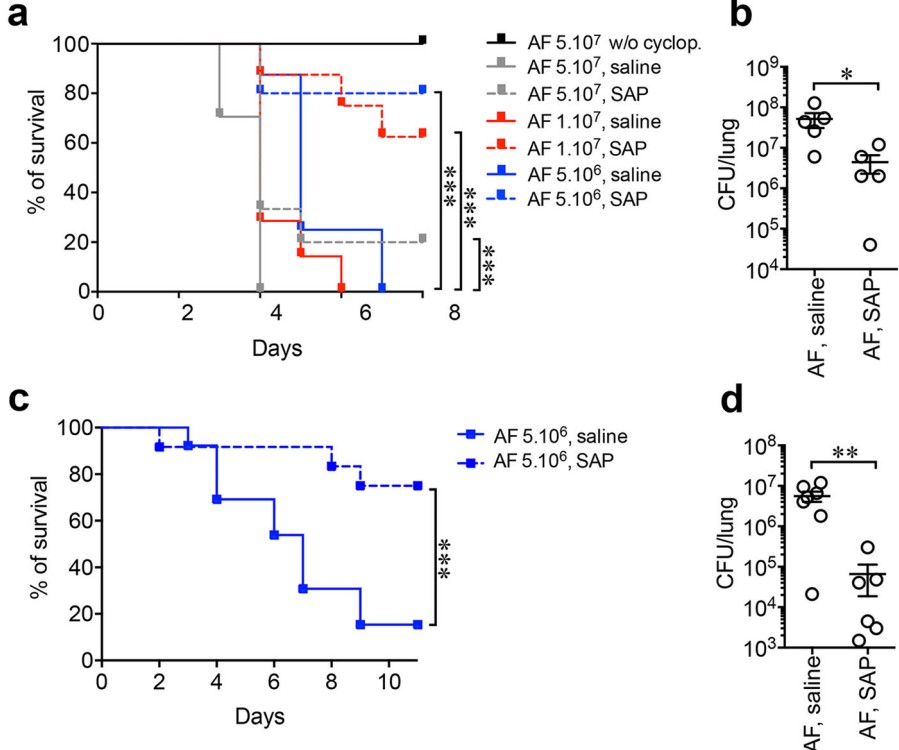

**Fig. 6 Therapeutic efficacy of SAP against invasive aspergillosis in immunosuppressed hosts. a** Percent survival of transiently immunosuppressed wt mice after i.t. injection with the indicated doses of AF conidia. The curve of untreated mice ($n = 3$) with cyclophosphamide is also shown. Human SAP (4 mg/kg) was i.p. injected at 2 h and 24 h after infection. AF $5 \times 10^7$, $n = 17$ mice; AF $5 \times 10^7 +$ SAP, $n = 15$ mice; AF $1 \times 10^7$, $n = 7$ mice; AF $1 \times 10^7 +$ SAP, $n = 8$ mice; AF $5 \times 10^6$, $n = 8$ mice; AF $5 \times 10^6 +$ SAP, $n = 10$ mice. An independent experiment shown. SAP-treated vs. saline, $P = 0.002$, AF $5 \times 10^7$; $P = 0.001$, AF $1 \times 10^7$; $P = 0.004$, AF $5 \times 10^6$ (Log-rank, Mantel-Cox test). **b** Lung CFU in mice treated with cyclophosphamide and infected with $1 \times 10^7$ AF conidia. $N = 5$ AF, $n = 5$ AF + SAP. Mean ± SEM of an independent experiment. *$P = 0.02$ (two-sided, Mann–Whitney $U$-test). **c, d** Effect of local administration of SAP on survival (**c**) and lung CFU (**d**) after i.t. injection of $5 \times 10^6$ in wt mice treated with saline (50 µl/mouse; $n = 13$) or human SAP (50 µg/50 µl/mouse; $n = 12$) 2 h after infection. ***$P = 0.003$ (Log-rank, Mantel-Cox test). **c** One experiment shown out of 2 performed with similar results. **d** Each spot corresponds to a single mouse. Mean ± SEM of an independent experiment. **$P = 0.008$ (two-sided, Mann–Whitney $U$-test).

**Table 1 Genotype distribution of *APCS* genotypes among cases of IPA and controls.**

| SNP rs# number | Alleles: status | Genotype, n (%)[a] | | | P-value[b] |
|---|---|---|---|---|---|
| | | A/A | A/a | a/a | |
| rs2808661 | G > A | | | | |
| Donor | IPA | 79 (71.2) | 28 (25.2) | 4 (3.6) | 0.04 |
| | Controls | 269 (72.3) | 99 (26.6) | 4 (1.1) | |
| Recipient | IPA | 76 (68.5) | 30 (27.0) | 5 (4.5) | 0.11 |
| | Controls | 276 (74.2) | 89 (23.9) | 7 (1.9) | |
| rs3753869 | C > A | | | | |
| Donor | IPA | 66 (59.5) | 37 (33.3) | 8 (7.2) | 0.04 |
| | Controls | 261 (70.2) | 99 (26.6) | 12 (3.2) | |
| Recipient | IPA | 72 (64.9) | 36 (32.4) | 3 (2.7) | 0.67 |
| | Controls | 213 (57.3) | 147 (39.5) | 12 (3.2) | |

*SNP single nucleotide polymorphism, IPA invasive pulmonary aspergillosis.*
[a]The major and minor alleles are represented by the first and second nucleotides, respectively.
[b]P-values were calculated using the cumulative incidence method and compared using Gray's test (two-sided). P-values are reported for the comparison between A/A + A/a and a/a.

*Apcs*$^{-/-}$ mice showed increased susceptibility to infection by *A. fumigatus*, a phenotype rescued by in vitro opsonization of fungi or in vivo administration of murine SAP. In vitro and in vivo observations suggest that complement activation following fungal recognition and phagocytosis of opsonized fungi are responsible of SAP-mediated resistance.

SAP has been shown to bind a variety of microbial species and to have opsonic activity[16,19,22–24,26], though under selected conditions,

inhibition of phagocytosis has been reported[26,54]. Recently, SAP was found to bind amyloid fibrils produced on the surface of *Candida albicans* and to dampen recognition by the innate immune system[55]. Here, we found that SAP binds *A. fumigatus* conidia and facilitates phagocytosis, and evidence suggests that this pathway underlies the role of this fluid phase PRM in resistance to this fungus.

The role of SAP in resistance to *A. fumigatus* is reminiscent of findings with the related molecule PTX3[38]. Genetic evidence in

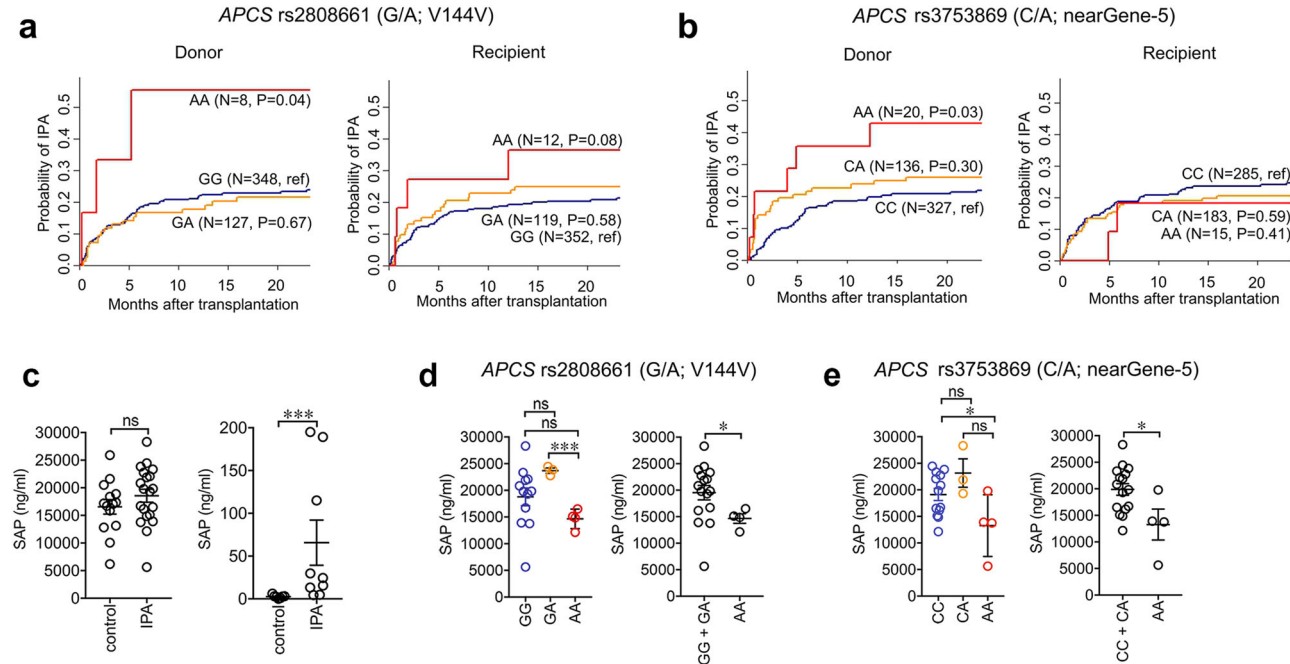

**Fig. 7 SNPs in *APCS* predispose stem-cell transplant recipients to IPA. a, b** Cumulative incidences of IPA according to *APCS* genotypes (IFIGEN cohort) are shown. **a** Cumulative incidence according to donor or recipient rs2808661 genotypes: donor GG (22.8%, n = 348, reference), GA (21.4%, n = 127; P = 0.67) and AA (55.6%, n = 8; P = 0.04), and recipient GG (20.3%, n = 352, reference), GA (25.0%, n = 119; P = 0.58) and AA (36.4%, n = 12; P = 0.08). **b** Cumulative incidence according to donor or recipient rs3753869 genotypes: donor CC (20.6%, n = 327, reference), CA (25.9%, n = 136; P = 0.30) and AA (42.9%, n = 20; P = 0.03), and recipient CC (23.4%, n = 285, reference), CA (20.4%, n = 183; P = 0.59) and AA (18.2%, n = 15; P = 0.41). Data were censored at 24 months, and relapse and death were competing events. **a, b** P-values are for Gray's test (two-sided). **c** Levels of SAP in sera (left) and BALFs (right) of hematological patients from the FUNBIOMICS cohort. Sera, controls, n = 14; IPA, n = 20. BALFs, n = 7; IPA, n = 9. Mean ± SEM. ***P = 0.0007 (two-sided, Mann–Whitney *U*-test). **d, e** Levels of SAP in serum of IPA patients with donor rs2808661 (**d**) and donor rs3753869 (**e**). Donor rs2808661, GG (n = 12), GA (n = 3), AA (n = 4) (**d**) and Donor rs3753869, CC (n = 13), CA (n = 3), AA (n = 4) (**e**) analyzed separately for the different genotype (**d, e**, left) or AA vs. GG + GA (**d**, right) and AA vs. CC + CA (**e**, right). AA vs. GA, ***P = 0.0006 (two-sided, unpaired *t*-test); AA vs. GG + GA, *P = 0.05 (two-sided, Mann–Whitney *U*-test); AA vs. CC, *P = 0.03 (two-sided, unpaired *t*-test); AA vs. CC + CA, *P = 0.02 (two-sided, unpaired *t*-test). **c–e**, Mean ± SEM.

---

**Table 2 Multivariate analysis of the association of *APCS* SNPs with the risk of IPA among transplant recipients.**

| Genetic/clinical variables | Adjusted HR[a] (95% CI) | P-value |
|---|---|---|
| Donor AA genotype in rs3753869 | 3.59 (1.54–8.78) | 0.012 |
| Unrelated donor | 1.98 (1.04–3.91) | 0.041 |
| Acute GVHD grades III–IV | 2.37 (1.29–3.55) | 0.006 |

Multivariate analyses were based on the subdistribution regression model of Fine and Gray. *HR* hazard ratio, *CI* confidence interval.
[a]Hazard ratios were adjusted for patient age and gender, type of transplantation, acute GVHD III–IV and antifungal prophylaxis. Only the variables remaining significant after adjustment are shown. The P-values are two-sided and no corrections for multiple testing were performed.

---

mice[2,38] and humans[39–47] suggests that PTX3 plays an essential role in resistance against selected pathogens, in particular *A. fumigatus*. SAP is produced in the liver whereas PTX3 is stored in neutrophil granules and produced by macrophages as well as by other cell types[1,2]. It is therefore likely that SAP and PTX3 exert through different molecular mechanisms[51] complementary functions systemically and in peripheral tissues in antifungal resistance.

In hematological patients, we found that BALF had substantially higher levels of SAP in IPA compared to controls, but not peripheral blood. Moreover, in patients undergoing stem cell transplantation, donor genetic variation was associated with increased susceptibility to IPA. These results raise the interesting possibility that a relevant source of SAP is represented by hematopoietic progenitor cells. The liver is the major dominant source of SAP[7,9,10]. However, local expression of SAP has been reported in atherosclerosis[56] and in fibrotic lesions[10]. By bioinformatics analysis, we found evidence of *APCS* gene expression in human myelomonocytic progenitors, myeloid cells in acute coronary syndromes and in sepsis and myeloid murine cells (Supplementary Fig. 12a–f). Moreover, *APCS* was expressed in monocytes in our own cohort of COVID-19 patients[57] as assessed by RNAseq (Supplementary Fig. 12f). Further studies are required to define the relevant extrahepatic cellular source of SAP in patients with IPA.

There are well-known risk factors that predispose to fungal disease, particularly in immunocompromised and severely ill patients, but these alone fail to convincingly explain the development of infection in all patients. Several studies have reported an expanding number of SNPs associated with IPA[40,43,58]. We now demonstrate that SNPs in *APCS* also contribute to IPA via molecular mechanisms influencing opsonisation and activation of the complement cascade in response to fungal infection. Although genetic variation in *APCS* has been evaluated in the context of other diseases, including type 2 diabetes and polyneuroendopathic amyloidosis[59], these are not acknowledged as relevant conditions predisposing to IPA. Our analyses have indeed confirmed a role for *APCS* SNPs in IPA that is independent from clinical risk factors often associated with the infection in the stem-cell transplantation setting, such as the development of severe acute GVHD. These data may become relevant in the clinical management of patients at-risk of IPA, since the identification of patients that may benefit the most from SAP-based therapies may be possible.

The frequency of *APCS* homozygous mutant genotypes associated with IPA is relatively low in the population. However, this fact further strengthens the critical role of these genetic variants as risk factors for IPA, since carriers of these genotypes reach cumulative incidence values of infection around 50%. Therefore, although rare in the general population, specific *APCS* genotypes are greatly enriched among IPA patients, suggesting a particularly relevant mechanism in the pathogenesis of infection, thus supporting a potential clinical applicability for screening donors, for patient management, e.g., to direct antifungal prophylaxis or to intensify fungal diagnostics in patients at higher genetic risk.

SAP has been shown to regulate matrix deposition by regulating macrophages, which play a key role as orchestrators of tissue remodeling[10,31–33]. These preclinical results paved the way to clinical development of SAP (PRM-151) as an anti-fibrotic agent[34,35]. Early Phase II results of PRM-151 in idiopathic pulmonary fibrosis are encouraging[36,37]. Current therapeutic limitations as well as concerns over the emergence of antifungal resistance are inspiring the search for novel host-directed therapies. The results reported here suggest that SAP may be explored as a therapeutic agent in *Aspergillus* infections, which represent a formidable clinical challenge.

## Methods

**Animals**. Wild-type (wt) C57BL/6J and C57BL/6NJ mice between 8 and 10 weeks of age were, respectively, purchased from Charles River Laboratories (Calco, Como, Italy) and from The Jackson Lab (Bar Harbor ME, US). $Apcs^{-/-}$ mice were generated as described[27]. $C1q^{-/-}$ mice were kindly provided by Prof. John Lambris (University of Pennsylvania, USA). $C3^{-/-}$, $Mbl1/2^{-/-}$ and $Fb^{-/-}$ mice were purchased from Jackson Lab (Bar Harbor ME, US). $Apcs^{-/-}$, $C1q^{-/-}$, $C3^{-/-}$, $Mbl1/2^{-/-}$ mice were on a C57BL/6J genetic background, whereas $Fb^{-/-}$ mice were on C57BL6/NJ. wt littermates were used as control for $Mbl1/2^{-/-}$, $C1q^{-/-}$ and $Fb^{-/-}$ mice. Co-housed wt were used as control for $C3^{-/-}$ mice. Mice were housed and bred in the SPF animal facility of Humanitas Clinical and Research Center in individually ventilated cages (light-dark cycle: from 7 a.m. to 7 p.m.; temperature and humidity: 20–24 °C and 40–60%). Procedures involving animals and their care were conformed to protocols approved by the Humanitas Clinical and Research Center (Rozzano, Milan, Italy) in compliance with national (4D.L. N.116, G.U., suppl. 40, 18-2-1992) and international law and policies (EEC Council Directive 2010/63/EU, OJ L 276/33, 22-09-2010; National Institutes of Health Guide for the Care and Use of Laboratory Animals, US National Research Council, 2011). The study was approved by the Italian Ministry of Health (approval n. 71/2012-B, issued on the 09/03/2012, 44/2015-PR issued 28/01/2015 and 742/2016-PR issued on the 26/07/2016). All efforts were made to minimize the number of animals used and their suffering.

**Population cohorts**. For the genetic association study, a total of 483 hematological patients of European ancestry undergoing allogeneic HSCT at Instituto Português de Oncologia, Porto, and at Hospital de Santa Maria, Lisbon (Portugal), were enrolled in the IFIGEN study between 2009 and 2015. The demographic and clinical characteristics of the patients are summarized in Supplementary Table 1. One hundred-eleven cases of probable/proven IPA were identified according to the 2008 criteria from the European Organization for Research and Treatment of Cancer/Mycology Study Group (EORTC/MSG)[60]. Twenty-one patients were excluded from the study based on the "possible" classification of infection. The levels of SAP were measured in clinical specimens from adult hematological patients enrolled in the FUNBIOMICS study[53] at the Leuven University Hospitals, Leuven (Belgium). Cases of probable/proven IPA were diagnosed as above[60]. The control group included patients with no evidence for the presence of *Aspergillus* spp. in the bronchoalveolar lavage (BALF) sample (negative culture, negative microscopy, and a galactomannan optical density index <0.5). Patients with "possible" disease were excluded from the study and no mold-active antifungals were administered by the treating physician(s) before sample collection. BALF and serum samples were collected using standard protocols, as described[53].

**Ethics statement**. Approval for the IFIGEN study was obtained from the Ethics Committee for Research in Life and Health Sciences (CEICVS) of the University of Minho, Portugal (no. 125/014), the Ethics Committee for Health of the Instituto Português de Oncologia - Porto, Portugal (no. 26/015), the Ethics Committee of the Lisbon Academic Medical Center, Portugal (no. 632/014), and the National Commission for the Protection of Data, Portugal (no. 1950/015). The FUNBIOMICS study was approved by CEICVS and the Ethics Committee of the University Hospitals of Leuven, Belgium (no.126/014). Written informed consent was obtained from the patient or a representative prior to collection of samples.

The study on COVID-19 patients was approved by the local Humanitas Clinical and Research Center Ethical Committee (authorization 233/20). This study was conducted as described[57] on a cohort of patients admitted to Humanitas Clinical and Research Center (Rozzano, Milan, Italy) with a laboratory-confirmed diagnosis of COVID-19.

In healthy donors, neutrophils and monocytes were isolated from peripheral blood upon approval by the Humanitas Research Hospital Ethical Committee (approval obtained on January 28th, 2016).

**Invasive pulmonary aspergillosis**. A clinical strain of *A. fumigatus* was isolated from a patient with a fatal case of pulmonary aspergillosis was kindly provided by Dr. Giovanni Salvatori (Sigma-tau, Rome, Italy). *A. flavus* (#MCV-C#1; ATCC® 204304) and *A. terreus* (Thom, #MF4833; ATCC® 20541) were purchased from ATCC®. The growth and culture conditions of *A. fumigatus*, *A. flavus* and *A. terreus* conidia were as described[38]. The *Candida albicans* strain CAF3-1 was kindly provided by Dr. Marina Vai (Dept. of Biotechnology and Biosciences, University of Milano-Bicocca). *C. albicans* blastospores were grown at 25 °C in rich medium [YEPD (yeast extract, peptone, dextrose), 1% (w/v) yeast extract, 2% (w/v) Bacto Peptone, and 2% (w/v) glucose] supplemented with uridine (50 mg/l) and the hyphae induced as described[61]. For intratracheal (i.t.) injection, mice were anesthetized with ketamine (100 mg/kg; i.p.) and xylazine (10 mg/kg i.p.). After surgical exposure, a volume of 50 μl PBS$^{2+}$, pH 7.4, containing $1 \times 10^8$ or $5 \times 10^7$ resting conidia (>95% viable, as determined by serial dilution and plating of the inoculum on Sabouraud dextrose agar) were delivered into trachea under direct vision using a catheter connected to the outlet of a micro-syringe (Terumo, Belgium). Survival to infection was daily monitored for 10d later. *C. albicans* blastospores ($5 \times 10^6$ in a final volume of 50 μl PBS) were intravenously (i.v.) injected and survival monitored for 20 days. Dying mice were euthanized after evaluation of the following clinical parameters: body temperature dropping, intermittent respiration, solitude presence, sphere posture, fur erection, non-responsive alertness, and inability to ascent when induced.

In experiments of in vivo phagocytosis, mice were i.t. injected with $5 \times 10^7$ heat-inactivated fluorescein isothiocynate (FITC)-labeled conidia and euthanized 4 h later. In rescue experiments, conidia ($1 \times 10^9$) were opsonized with murine recombinant SAP (50 μg/ml; lot. #MPX1518061; R&D Systems) for 1 h at r.t. in PBS, pH 7.4, containing 0.01% (vol/vol) Tween-20® (Merck-Millipore). After washing of unbound protein, a volume of 50 μl ($5 \times 10^7$ conidia) was i.t. injected. Alternatively, a single i.v. injection of murine SAP (50 μg/mouse; lot. #MPX1518061; R&D Systems) previously dialyzed in PBS (pH 7.4) in order to eliminate sodium azide was given after 2 h i.t. injection of conidia ($5 \times 10^7$). In therapy experiments, immunosuppression was induced by i.p. injection of 150 mg/kg cyclophosphamide (150 μl per mouse of 20 mg/ml solution) 2 days before infection and indicated doses of conidia were i.t. injected. Native human SAP (lot. #2787319; Merck-Millipore) was dialyzed in PBS (pH 7.4) and i.p. injected at the dose of 4 mg/kg at 2 and 24 h after infection. In experiments of local therapy, 50 μg human SAP (1 mg/ml; 50 μl in PBS) were i.t. administred 2 h after infection.

**BALFs collection and analysis**. BALFs were performed with 1.5 ml PBS, pH 7.4, containing protease inhibitors (Complete tablets, Roche Diagnostic; PMSF, Sigma-Aldrich/Merck) and 10 mM EDTA (Sigma-Aldrich/Merck) with a 22-gauge venous catheter. BALFs were centrifuged, and supernatants were collected for quantification of cytokines as described below. After erythrocyte lysis with ACK solution (pH 7.2; $NH_4Cl$ 0.15 M, $KHCO_3$ 10 mM, EDTA 0.1 mM), cells were resuspended in PBS, pH 7.4, containing 10 mM EDTA and 1% heat-inactivated fetal bovine serum (FCS; Sigma-Aldrich/Merck), stained with LIVE/DEAD™ Fixable Violet Dead Cell Stain Kit (ThermoFisher Scientific-Molecular Probes) or Zombie Aqua™ Fixable Viability Kit (Biolegend, US), and analyzed as reported in Supplementary Fig. 13 by BD FACS LSRFortessa™ Flow Cytometer (BD Biosciences) and BD FACSDiva™ Software (version 6.2) with the following specific antibodies: peridinin chlorophyll protein complex (PerCP)- or brilliant violet (BV) 650-labeled anti-CD45 (#30-F11, $IgG_{2B}$; used at 4 μg/ml; Cat. No. 103130, Biolegend, and Cat. No. 563410, BD Biosciences, respectively); FITC- or phycoerythrin (PE)-CF594-labeled anti-Ly6G (#1A8, $IgG_{2A}$; used at 4 μg/ml; Cat. No. 11-9668-82, eBioscience-Invitrogen, and Cat. No. 562700, BD Biosciences, respectively); allophycocyanin (APC)-cyanine (Cy)7- or BV421-labeled anti-CD11b (#M1/70, RUO, $IgG_{2B}$; used at 1 μg/ml; Cat. No. 557657 and Cat. No. 562605, BD Biosciences); BV421-labeled anti-Ly6C (#AL-21, RUO, IgM; used at 1.6 μg/ml; Cat. No. 562727, BD Biosciences), APC-labeled anti-CD11c (#HL3, $IgG_1$; used at 1.6 μg/ml; Cat. No. 550261, BD Biosciences) and PE-Cy7-labeled anti-F4/80 (#BM8, $IgG_{2A}$; used at 1.6 μg/ml; Cat. No. 123110, Biolegend). At least 10,000 events of CD45$^+$ cells were considered as cut-off for the FACS analysis.

**Lung homogenates and analysis**. Lungs were removed 16 h after infection and homogenized in 1 ml PBS, pH 7.4, containing 0.01% (vol/vol) Tween-20® (Merck-Millipore) and protease inhibitors. Samples were serially diluted 1:10 in PBS and plated on Sabouraud dextrose agar for blinded CFU counting. For lysate preparation, lungs were collected at 4 h and homogenized in 50 mM Tris-HCl, pH 7.5, containing 2 mM EGTA, 1 mM PMSF, 1% Triton X-100 (all from Sigma-Aldrich/Merck), and complete protease inhibitor cocktail. Total proteins were measured by

DC Protein Assay, according to manufacturer's instructions (Bio-Rad Laboratories). Western blot analysis for C3 was performed after loading 10 μg of lung protein extracts on sodium dodecyl sulfate polyacrylamide gel electrophoresis (SDS-PAGE). The goat polyclonal anti-C3 (diluted 1:3000; Merck-Millipore, Cat. No. 204869) and HRP-conjugated donkey anti-goat IgG (diluted 1:5000; R&D Systems, Cat. No. HAF-109) were used. The monoclonal anti-vinculin (#hVIN-1; used at 0.5 μg/ml; Sigma-Aldrich/Merck, Cat. No. V9264) was used as loading control. C3d bands were quantified by Fiji-ImageJ (NIH, Bethesda USA; version 2.1.0/1.53.c) as a ratio of mean gray intensity values of each protein relative to vinculin bands.

**Cells and in vitro phagocytic activity**. Phagocytosis assay in whole blood of *A. fumigatus* conidia by mouse and human neutrophils was performed as described[51]. Briefly, conidia ($1 \times 10^9$) were labeled (1 h, r.t.) with FITC (Sigma-Aldrich/Merck; 5 mg/ml in DMSO), and eventually opsonized (1 h, r.t) with murine or human SAP (100 μg/ml; 1.1 μM) and PTX3 (50 μg/ml; 1.1 μM). An amount of $1 \times 10^7$ FITC-conidia were added to 200 μl of mouse whole blood (collected with heparin from cava vein) and incubated for 1 and 20 min at 37 °C in an orbital shaker. Samples were immediately placed on ice and ACK lysis solution was added to lyse erythrocytes. Murine neutrophils, monocytes, macrophages and dendritic cells were analyzed by BD FACS LSRFortessa™ Flow Cytometer (BD Biosciences) as above described and indicated in Supplementary Fig. 13, and frequency and/or mean fluorescence intensity (MFI) of FITC+ neutrophils and/or CD11b expression were reported.

Human neutrophils were isolated from fresh whole blood of healthy volunteers through separation from erythrocytes by 3% dextran (GE Healthcare Life Sciences) density gradient sedimentation followed by Ficoll-Paque PLUS (GE Healthcare Life Sciences) and 62% Percoll® (Sigma-Aldrich/Merck) centrifugation as previously described[51]. Blood monocytes were obtained by Ficoll-Paque PLUS (GE Healthcare Life Sciences) and 46% Percoll® (Sigma-Aldrich/Merck). Purity, determined by FACS analysis on forward scatter/side scatter parameters, was routinely > 98%. $1 \times 10^5$ cells were incubated for 1 and 30 min at 37 °C in 50 μl RPMI-1640 medium with 5 and/or 10% normal human serum (NS) or complement depleted sera, optionally with autologous human plasma-heparin, and $1 \times 10^6$ FITC-labeled *A. fumigatus* conidia. When indicated, cells were resuspended in 10% normal plasma-heparin from healthy donors depleted from IgGs or SAP as described below. Cells were transferred on ice and, after washing with PBS, pH 7.4, containing 10 mM EDTA and 0.2% bovine serum albumin (BSA; Sigma-Aldrich/ Merck), FITC fluorescence in neutrophils (defined as FSC-Ahigh/SSC-Ahigh) and monocytes (defined as FSC-Adim/SSC-Adim) was analyzed by BD FACS Canto™ II Flow Cytometer (BD Biosciences) and BD FACSDiva™ Software (version 6.1.1). NS and sera depleted of C3, C1q, MBL, FB and FH were all obtained from CompTech (Complement Technology, Inc., USA). In blocking experiments, neutrophils were-preincubated for 1 h on ice in the presence of following mouse IgG$_1$ (used at 10 μg/ml/$1 \times 10^6$ cells): anti-CD16 (FcγRIII) (#3G8; BioLegend, Cat. No. 302002); anti-CD32 (FcγRII) (#FUN-2; BioLegend, Cat. No. 303202), or anti-CD32 (FcγRII) (#AT10; Merck-Millipore, Cat. No. MABF925); anti-CD64 (FcγRI) (#10.1; eBioscience-ThermoFisher, Cat. No. 14-0649-82); anti-M-ficolin (#036-051 1; Santa Cruz Biotechnology, Cat. No. sc-80486); L-ficolin (#FCN219; Santa Cruz Biotechnology, Cat. No. sc-80484); irrelevant mouse IgG$_1$ (#MG1-45, BioLegend, Cat. No. 401402, or #P3.6.2.8.1, ThermoFisher Scientific, Cat. No. 16-4714-82). All these antibodies were previously dialyzed in PBS, pH 7.4.

Bone marrow-derived (BM) cells were isolated from femurs and tibiae and cultured at density of $1.5 \times 10^6$ cells/ml in RPMI-1640 medium supplemented with 10% heat-inactivated Fetal Bovine Serum (FBS; Sigma-Aldrich/Merck), 2mM L-Glutamine, 1% Pen/Strept for 24 h. Non-adherent cells were then harvested and cultured with 20 ng/ml of murine recombinant M-CSF (PeproTech) to obtain macrophages. Cells were washed and medium replaced at day 3. BM macrophages ($1 \times 10^6$ cells/ml) were stimulated on day 6 with heat-inactivated *A. fumigatus* conidia (ratio 1:5, cell:conidia) or 10 ng/ml LPS (Sigma-Aldrich/Merck) in the presence of 10% freshly isolated serum obtained from wt and *Apcs*$^{-/-}$ mice to preserve functions of complement. Cell supernatants were collected (24 h) for cytokine measurement.

**Complement deposition on *Aspergillus fumigatus***. A volume of 10 μl PBS, pH 7.4, containing $1 \times 10^7$ conidia eventually opsonized with murine SAP (100 μg/ml per $1 \times 10^9$ conidia, 1 h at r.t.) was incubated (37 °C) in round bottom wells of Corning® 96-well polypropylene microplates for the indicated time points with 20 μl mouse plasma-heparin or 20 μl human NS, complement depleted sera (diluted in PBS at 10% and 30%). When indicated, plasma-heparin obtained from healthy donors depleted of IgGs or SAP (diluted in PBS at 10%) was used. Complement deposition was blocked by addition of EDTA (10 mM final concentration) and by cooling in ice. After centrifugation (1450 x g, 10 min at 4 °C), when indicated, supernatant was collected for mouse C5a measurement by ELISA. Conidia were washed and incubated (1 h, at 4 °C) with PBS, pH 7.4, 2 mM EDTA, 1% BSA containing the following primary antibodies: goat anti-human/mouse C3 and activation fragments (diluted 1:5000; Merck-Millipore, Cat. No. 204869); rat anti-mouse C1q (IgG$_1$, #7H8; used at 1 μg/ml; HyCult® Biotech, Netherlands, Cat. No. HM1044); rabbit anti-human C5b-C9 (MAC) (diluted 1:2000; Complement Technology, Inc., Cat. No. A227); or correspondent irrelevant IgGs. Conidia were

then incubated (1 h, at 4 °C) with Alexa Fluor® 488 and 647-conjugated species-specific cross-adsorbed detection antibodies (used at 2 μg/ml; ThermoFisher Scientific-Molecular Probes, Cat. No. A-11055, A-48272, A-21206) and analyzed by BD FACS Canto™ II Flow Cytometer (BD Biosciences) using forward and side scatter parameters to gate at least 8000 conidia. After each step, conidia were extensively washed with PBS, pH 7.4, 2 mM EDTA, 1% BSA. Results were expressed as frequency of conidia showing fluorescence compared with irrelevant controls and as geometric conidia MFI.

**Fungal viability assay**. Effect of SAP on *A. fumigatus* killing was evaluated by a resazurin-based cell viability assay. A volume of 10 μl PBS, pH 7.4, containing $1.5 \times 10^5$ conidia eventually opsonized with human SAP (100 μg/ml per $1 \times 10^9$ conidia, 1 h at r.t.) was placed into sterile round bottom Corning® 96-well polypropylene microplate and incubated for 1 and 30 min at 37 °C with 20 μl of human serum and different complement component-depleted sera (30%). After incubation, plates were immediately cooled on ice and cold-centrifuged (1450 x g, 10 min at 4 °C), and then supernatant removed. Conidia not incubated with serum were used as a negative control. Conidia treated with the fungicide drug Posaconazole (POC; 1 μM) were considered as positive control in the assay. Preparation of Alamar-Blue™ Cell Viability Reagent and test was performed according with manufacture's instructions (ThermoFisher Scientific-Invitrogen). A volume of 100 μl AlamarBlue™ solution (10 μl of AlamarBlue™ reagent and 90 μl of Sabouraud dextrose broth) was added to each well. After 17 h reaction at 37 °C, fluorescence (excitation/emission at ≈530–560/590 nm) intensity was measured by microplate reader Synergy™ H4 (BioTek, France). Results represent ratio of fluorescence intensity values relative to those measured in negative controls. The actual killing of fungi was controlled as CFU count.

**Proteins**. Recombinant murine SAP from mouse myeloma cell line NSO was used (lot. #MPX1518061; R&D Systems). Native SAP from human serum was purchased by Merck-Millipore (lot. #2787319). Recombinant murine PTX3 was purified from supernatant of Chinese hamster ovary cells (ATCC®-CRL-12023) stably expressing the protein, as described previously[51]. Purity of the recombinant protein was assessed by SDS-PAGE followed by silver staining. Proteins contained <0.125 endotoxin units/ml as checked by the Limulus amebocyte lysate assay (BioWhittaker®, Inc.). SAP levels were measured in mouse plasma by ELISA (DuoSet ELISA; R&D Systems) after collection of blood from the cava vein. Murine MPO was measured in BALFs by ELISA (DuoSet ELISA; R&D Systems). Murine CXCL1, CXCL2, TNF-α and IL-6 were measured in supernatants by ELISA (DuoSet ELISA; R&D Systems) according with manufacturer's instructions. Murine C5a was measured either in plasma-heparin or in BALFs previously stored at −80 °C by DuoSet ELISA (R&D Systems) maintaining EDTA (10 mM) throughout the assay in order to stop the activation of the complement cascade. Human SAP was measured in sera and BALFs of patients by ELISA (lot. #P105304, DuoSet ELISA; R&D Systems) according with manufacturer's instructions.

**IgGs and SAP depletion**. Plasma-heparin of healthy volunteers was collected in BD Vacutainer® and maintained (1 h) on ice during procedures of depletion to avoid the activation of the complement. IgG depletion was obtained by passing human plasma-heparin (2 ml) on protein-G Sepharose™ Fast Flow (GE Healthcare, Sweden) column, as indicated by manufacturer's instructions. Depletion of SAP was obtained by passing plasma heparin (2 ml) of each donor through different columns of protein-G Sepharose™ Fast Flow (GE Healthcare, Sweden) previously cross-linked with a purified rabbit polyclonal anti-SAP (1 mg/1 ml of resin) (Merck-Millipore, Cat. No. 565191). Dimethyl pimelimidate (Sigma-Aldrich/ Merck) was used as cross-linking reagent. Bound proteins were eluted with 0.1 M Glicine-HCl pH 2.8, concentrated with Vivaspin® 10,000 MWCO (Sartorius, Germany) and titrated with the Pierce™ Coomassie (Bradford) protein assay kit (ThermoFisher Scientific). The actual depletion of SAP and IgGs was evaluated by Western blot analysis after loading 1 μl/lane plasma-heparin on SDS-PAGE. Proteins eluted from the anti-SAP column were also loaded to ascertain its capture. The mouse monoclonal anti-SAP (used at 0.5 μg/ml; #910119; R&D Systems, Cat. No. MAB1948) and HRP-conjugated sheep anti-mouse IgG (diluted 1:3000; GE Healtcare, Sweden, Cat. No. NA931V) or a goat anti-human IgG (used at 1 μg/ml; Jackson ImmunoResearch, Cat. No. 109-006-097) and HRP-conjugated anti-goat IgG (diluted 1:5000; R&D Systems, Cat. No. HAF-109) were used to detect, respectively, SAP and IgGs. The different intensity of SAP (or IgG) bands between lanes Donor A–E vs. Donor A–E/IgG depleted (or SAP depleted) are due to steric hindrance of antibody recognition, having SAP and IgG low chains similar molecular weight.

**Binding of SAP**. *Aspergillus* conidia ($1 \times 10^8$ CFU) were cultured 4 and 16 h under static condition in Sabouraud medium to, respectively, allow conidia swelling and germination. *C. albicans* blastospore, yeast and hyphae ($1 \times 10^7$ CFU) were obtained as described[61]. After washing with PBS$^{2+}$, pH 7.4, containing 0.01% (vol/vol) Tween-20® (Merck-Millipore), resting and swollen conidia or hyphae were incubated (1 h, r.t.) with biotin-labeled murine SAP (R&D Systems) at concentrations ranging from 0.1 to 10 μg/ml in PBS$^{2+}$, pH 7.4, containing 2% BSA (Sigma-Aldrich/Merck). In competition experiments, human SAP or CRP (50 μg/

ml; lot. #B68019; Merck-Millipore) or murine PTX3 (50 µg/ml) were further added. After extensive washing, samples were incubated (30 min, r.t.) with Alexa Fluor® 647-conjugated streptavidin (diluted 1:1000; ThermoFisher Scientific-Molecular Probes, Cat. No. S-32357) and binding was evaluated by FACS as frequency and MFI and visualized by confocal microscopy as described. In some experiments, a rat monoclonal anti-SAP (IgG$_{2A}$, #300103; used at 5 µg/ml; R&D Systems, Cat. No. MAB2558) and an Alexa Fluor® 647-conjugated goat anti-rat IgG (diluted 1:500; ThermoFisher Scientific-Molecular Probes, Cat. No. A21247) were also used. CRP was cross-linked with BS3 (Bis[sulfosuccinimidyl]suberate; Thermo Fisher Scientific) according to the manufacturer's instructions.

**Microscopy.** Five micrometer cryostat sections of mouse lungs injected with $5 \times 10^7$ heat inactivated FITC-labeled conidia (4 h) were incubated in 5% of normal donkey (Sigma-Aldrich/Merck) serum, 2% BSA (Sigma-Aldrich/Merck), 0.1% Triton X-100 (Sigma-Aldrich/Merck) in PBS$^{2+}$, pH 7.4, (blocking buffer) for 1 h at room temperature. Sections were incubated with the following primary antibodies diluted in blocking buffer for 2 h at r.t.: rabbit polyclonal anti-SAP (diluted 1:500; Merck-Millipore, Cat. No. 565191); rat monoclonal Ab anti-C3 (C3b/iC3b/C3d) (IgG$_{2A}$, #2/11; used at 5 µg/ml; Hycult® Biotech, Cat. No. HM1065) previously conjugated with Alexa Fluor® 647 through the Antibody Labeling kit (Thermo-Fisher Scientific-Molecular Probes) and BV421-labeled anti-Ly6G (IgG$_{2A}$, #1A8; used at 1 µg/ml; BD Biosciences, Cat. No. 562737). Sections were then incubated for 1 h with an anti-rabbit IgG Alexa Fluor® 532-conjugated detection antibody (used at 1 µg/ml; ThermoFisher Scientific-Molecular Probes, Cat. No. A-11009). After each step, sections were washed with PBS$^{2+}$, pH 7.4, containing 0.01% (vol/vol) Tween-20® (Merck-Millipore). Correspondent IgG isotype controls (BV421- and Alexa Fluor® 647-labeled rat IgGs from BD Biosciences, respectively, Cat. No. 562602 and 557690, and a purified homemade rabbit IgG) were used. Lungs obtained from Apcs$^{-/-}$ mice were used as control. Sections were mounted with the antifade medium FluorPreserve® Reagent (Merck-Millipore) and analyzed in a sequential scanning mode with a Leica TCS SP8 confocal microscope at Airy Unit 1 (pinhole aperture 95.5 µm) with an oil immersion lens HC PL APO 63X (N.A. 1.4) equipped with Leica Application Suite X software (LASX; version 3.5.5.19976). Images of SAP binding to resting or germinated conidia were obtained after z-stack acquisition using same instrument parameters and image deconvolution by Huygens Professional software (Scientific Volume Imaging B. V.; version 19.10) and presented as medium intensity projection (MIP).

**SNP selection and genotyping.** Genomic DNA was isolated from whole blood of patients enrolled in the IFIGEN study using the QIAcube automated system (Qiagen). SNPs were selected based on their ability to tag surrounding variants with a pairwise correlation coefficient $r^2$ of at least 0.80 and a minor allele frequency ≥5% using publically available sequencing data from the Pilot 1 of the 1000 Genomes Project for the CEU population. Supplementary Table 3 reports sequences of the primers used in the study. Genotyping was performed using KASPar assays (LGC Genomics) in an Applied Biosystems 7500 Fast Real-Time PCR system (Thermo Fisher Scientific), according to the manufacturer's instructions. Mean call rate for the SNPs was > 98%. Quality control for the genotyping results was achieved with negative controls and randomly selected samples with known genotypes.

***In silico* data analysis.** Publicly available expression data relative to human and murine myeloid cells were retrieved from the open access platform Gene Expression Omnibus archive (GEO). Data relative to human mononuclear BM cells were retrieved from the GEO data series GSE42519[62,63] based on Affymetrix Human Genome U133 Plus 2.0 Array. In this experiment, mononuclear BM cells from healthy donors were enriched for CD34$^+$ and hematopoietic stem and progenitor cells were sorted by FACS. Hematopoietic stem cells (HSCs) and hematopoietic progenitors (HPCs) with the following immunophenotypes were purified as described using a BD FACS Aria I$^{TM}$ cell sorter (BD Bioscience): HSCs, Lin$^-$ CD34$^+$ CD38$^-$ CD90$^+$ CD45RA$^-$; multipotent progenitors, Lin$^-$, CD34$^+$, CD38$^-$ CD90$^-$ CD45RA$^-$; common myeloid progenitors, Lin$^-$ CD34$^+$ CD38$^+$ CD45RA$^-$ CD123$^+$; megakaryocyte-erythroid progenitors, Lin$^-$ CD34$^+$ CD38$^+$ CD45RA$^-$ CD123$-$; granulocyte-macrophage progenitors, Lin$^-$ CD34$^+$ CD38$^+$ CD45RA$^+$ CD123$^+$. BM populations representing early and late promyelocytes, myelocytes, metamyelocytes, band cells, and BM polymorphonuclear cells were isolated by flow cytometry-based cell sorting: early promyelocytes, Lin$^-$ FSC$^{hi}$ SSC$^{int}$ CD34$^-$ CD15$^{int}$ CD49d$^{hi}$ CD33$^{hi}$ CD11b$^-$ CD16$^-$; late promyelocytes, Lin$^-$ FSC$^{hi}$ SSC$^{hi}$ CD34$^-$ CD15$^{hi}$ CD49d$^{hi}$ CD33$^{hi}$ CD11b$^-$ CD16$^-$; myelocytes, Lin$^-$ FSC$^{hi}$ SSC$^{hi}$ CD34$^-$ CD15$^{hi}$ CD49d$^{hi}$ CD33$^{hi}$ CD11b$^{hi}$ CD16$^-$; metamyelocytes, Lin$^-$ FSC$^{hi}$ SSC$^{hi}$ CD34$^-$ CD15$^{hi}$ CD49d$^-$ CD33$^-$ CD11b$^{hi}$ CD16$^-$; band cells, Lin$^-$ FSC$^{hi}$ SSC$^{hi}$ CD34$^-$ CD15$^{hi}$ CD49d$^-$ CD33$^-$ CD11b$^{hi}$ CD16$^{int}$; polymorphonuclear cells, Lin$^-$ FSC$^{hi}$ SSC$^{hi}$ CD34$^-$ CD15$^{hi}$ CD49d$^-$ CD33$^-$ CD11b$^{hi}$ CD16$^{hi}$. Data relative to monocytes and macrophages derived from patients affected by acute coronary syndrome were obtained from the GEO data series GSE11430[64] based on Affymetrix Human Genome U133 Plus 2.0 Array. In this study, blood samples were obtained from patients with symptoms of acute coronary syndrome who had undergone coronary angiography and who had one stenosis > 50% diagnosed in at least one major coronary artery. RNA samples were extracted from monocyte and monocyte derived macrophages. Data relative to human neutrophils were obtained from the GEO data series GSE3037[65] based on Affymetrix Human Genome U133A Array. Peripheral blood neutrophils were isolated from septic patients and treated in vitro with LPS or HMGB1. Since no raw data were available, data were processed starting with the gene expression matrix. Expression data of murine macrophage were obtained from the GEO data series GSE2935[66]. Mouse macrophage cultures were obtained from PBMCs isolated from wt mice inoculated with Sendai virus (SeV) or UV-inactivated SeV (UV-SeV). Expression was evaluated with Affymetrix Murine Genome U74A Version 2 Array. Since no raw data were available, data were processed starting with the gene expression matrix. Data of murine neutrophils were obtained from the GEO series GSE25211[67] based on Illumina MouseRef-8 v2.0 expression beadchip array. In this study, transcriptional differences of neutrophils isolated from wt and Il-1r1$^{-/-}$ mice after treatment with Ikkβ inhibitors were measured by Illumina MouseRef-8 v2.0 expression beadchip array. All microarray transcriptional data were processed starting with raw files when available, using R statistical environment (version 3.5.2) with the R packages "Biobase" version 2.42.0), "GEOquery" (version 2.50.5). Normalization, probes annotation, gene summarization and differential expression significance assessments were performed with the R package Limma (version 3.38.3). Bulk RNA-Seq data of human peripheral monocytes of COVID-19-infected patients and healthy controls were obtained from the GEO series GSE160351[57]. Associated transcriptomic data of human peripheral neutrophils derived from the same healthy donor samples were generated using same sequencing technology and protocols[57]. Briefly, RNA from neutrophils isolated from whole blood by negative selection using MACSxpress® Whole Blood Neutrophil Isolation Kit (Miltenyi Biotec, Germany) was purified with the Direct-zol RNA Microprep or Miniprep Kits (Zymo Research), according to manufacturer's instructions. Total-RNA-sequencing library preparation was performed starting from 1 ng of total-RNA with the SMART-Seq Stranded Kit (Clontech-Takara). Libraries obtained were qualitatively assessed by using TapeStation 4200 (Agilent) and quantified by Qubit Fluorimeter (Thermofisher). Afterwards, they were multiplexed in equimolar pools and sequenced on a NextSeq-550 Illumina Platform generating at least 80 million 75bp-PE reads per sample. Raw RNA-Seq data of human peripheral neutrophils derived from three healthy donors are deposited and publicly available under the GEO series GSE163533. Raw files were aligned and quantified with STAR (version 2.6.1) on the GRCh38 genome guided by GENCODE annotation (version 33). Gene summarized counts were processed in R, genes whose expression was minor than two reads were removed while the remaining portion was "vs. normalized" with the R package DESeq2[68] (version 1.22.1). Plots were rendered with the R library "ggplot2" (version 3.3.2).

**Statistic.** Student's *t*-tests were performed after data were confirmed to fulfill the criteria of normal distribution and equal variance. Otherwise Mann–Whitney test was applied. Log-rank (Mantel-Cox) test was performed for comparison of survival curves. Statistical significance of multivariate frequency distribution between groups was also analyzed by Fisher's Exact test. Analyses were performed with GraphPad Prism version 6 or 7c software. The probability of IPA according to *APCS* genotypes was determined using the cumulative incidence method and compared using Gray's test. Cumulative incidences of infection at 24 months were computed with the cmprsk package for R version 2.10.1, with censoring of data at the date of last follow-up visit and relapse and death as competing events. All clinical and genetic variables achieving a *P*-value ≤ 0.15 in the univariate analysis were entered one by one in a pairwise model together and kept in the final model if they remained significant (*P* < 0.05). Multivariate analysis was performed using the subdistribution regression model of Fine and Gray with the cmprsk package for R version 2.10.1.

**Reporting summary.** Further information on research design is available in the Nature Research Reporting Summary linked to this article.

## Data availability

All data needed to evaluate the conclusions in the paper are present in the paper and/or in the Supplementary Information and Data Source File. Additional data related to this paper may be requested from the authors. Source data are provided with this paper.

## Code availability

Bioinformatic methods used for in silico analyses were based on publicly available software, custom scripts might be requested to the authors. Microarray-based computational analyses were performed on publicly available datasets derived from the Gene Expression Omnibus (GEO) platform under the accession ID GSE42519, GSE11430, GSE3037, GSE2935, GSE25211, retrieved following these links: RNA-Seq data were derived from in-house performed sequencing experiments and are publicly available under the accession ID GSE160351 and GSE163533.

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

## Acknowledgements

The contribution of the European Commission (ERC project PHII-669415; FP7 project 281608 TIMER; ESA/ITN, H2020-MSCA-ITN-2015-676129), Ministero dell'Istruzione, dell'Università e della Ricerca (MIUR) (project FIRB RBAP11H2R9), Associazione Italiana Ricerca sul Cancro (AIRC IG-19014 and IG-21714, AIRC 5 × 1000 −9962 and −21147), the Italian Ministry of Health, the Northern Portugal Regional Operational Programme (NORTE 2020), under the Portugal 2020 Partnership Agreement, through the European Regional Development Fund (ERDF) (NORTE-01-0145-FEDER-000013 and NORTE-01-0145-FEDER-000023), the Fundação para a Ciência e Tecnologia (FCT) (UIDB/50026/2020, UIDP/50026/2020, PTDC/SAU-SER/29635/2017, PTDC/MED-GEN/28778/2017, CEECIND/04058/2018 and CEECIND/03628/2017), the European Union's Horizon 2020 research and innovation program under grant agreement no. 847507 and the "la Caixa" Foundation (ID 100010434) and FCT under the agreement LCF/PR/HR17/52190003 is gratefully acknowledged.

## Author contributions

A.D. contributed to the experimental design and conducted experiments, in particular animal model of invasive aspergillosis and in vitro experiments of binding and phagocytosis, and drafted the manuscript. R.P., I.L., Re.P. performed in vivo experiments. C.C. performed the genotyping study and statistical analysis. S.N.M., performed the *in silico* analyses. F.S.C., E.M. performed flow cytometry. J.F.L., A. Campos, T.M., K.L., J.M., A. Carvalho provided the patient samples and information in the context of the IFIGEN and FUNBIOMICS consortiums. A. Carvalho also participated in the drafting of the manuscript. F.P. prepared fungal culture. T.S. performed confocal microscopy. A.I., J.D.L., B.B. contributed to the scientific discussion of the project. C.G., M.B. contributed to the scientific discussion and participated in the drafting of the manuscript. A.C. conceived study of *Apcs* genotyping in patients and identified the relationship between genetic variation of *Apcs* and susceptibility to IPA and participated in drafting the manuscript. A.M. played a key role in designing and supervision of the study and drafted the final version of the paper.

## Competing interests

A.M. and A.D. are inventors of a patent on SAP (WO2020127471). The other authors do not report competing interests.
