## [Peer Review File · Nature Communications]

Reviewers' comments:

Reviewer #1 (Anti-fungal immunity, genomics)(Remarks to the Author):

The paper by Doni and colleagues describes the role of the pentraxin APCS in opsonisation - mediated defence against fungal infection. In many ways these results follow previous work on PTX3 which was also shown to be an opsonisation protein for *Aspergillus* that led to disease susceptibility in KO mice – a phenotype that was also reversed by pre-opsonisation of spores with the protein (Nature. 420 (6912): 182–6). Here the authors demonstrate that APCS deficiency results in susceptibility to *Aspergillus fumigatus* infection in an immunocompromised mouse model and that resistance to infection can be restored by addition of exogenous APCS to the pathosystem. They also present evidence that variants in the gene are associated with invasive disease in humans.

The work has been conducted by a world-class international team and is easily of a high enough standard for inclusion in Nature Communications. This is an important finding and has implications in understanding and treating fungal invasive disease. The paper is generally very well written, however there are a number of gaps in the paper that need to be addressed – in particular the paper contains no macrophage or other cell - type data, instead concentrating on neutrophils and there is no data on the function of the APCS variants in human cells. This data either needs to be included or its omission needs to be clearly justified before publication can be recommended.

Specific comments to be addressed:

1 - The correct current gene name for SAP/PTX2 is APCS – this should be used throughout.

2 - The section on human genetics needs considerable clarification and inclusion of data regarding numbers of individuals in the study that carried each variant, considerations of variant frequency in the general population and effect of the SNP on the protein. There are 483 individuals in the cohort of which 111 have IPA - how many carry the heterozygous/homozygous variants? This would allow an estimate of the importance of variants in this gene in disease. For Figure 5 the numbers of patients in each group are not clearly stated and need to be included in Figure 5 or figure legend – it certainly looks as if there are very few individuals in the homozygous variant groups which might account for the marginal p-values. A table in supplementary data would also be sufficient.

The prevalence of the variants in the study population needs to be discussed in the context of the consequence of carrying the SNP and in discussion of the prevalence of such variants in the general population – eg if the MAF for rs2808661 is 0.1 then homozygous variants will be rare in the population. Rs3753869 is more common and so on.

3 - It is not clear whether therapeutic potential can be realised easily. It is very encouraging that the protein is in human trials but some mention of how this would transfer to IPA is needed – how common do the authors really think this mechanism is in susceptibility to IPA? For example allele frequencies for some SNPs in the population seem low and the mechanism might not be particularly common in IPA. Are the authors proposing a diagnostic or therapeutic avenue, or both? Is the proposal to screen donors for presence of the variant or to treat individuals already infected with injections of the protein? This could form an

interesting and useful part of the discussion rather than the vague statements provided.

4 - There is a strong focus on neutrophil responses in the pathosystem. I certainly agree that neutrophils are a major player in response to *Aspergillus* - and there are clear and credible results from the neutrophil work in this paper – but what about the other immune cells involved in the response? Is the effect of APCS opsonisation only on neutrophil phagocytosis or directly on neutrophil recruitment or does it for example stimulate DCs or Macrophages to emit neutrophil recruiting chemokines? The authors need to carefully consider this and justify their tight focus on neutrophils more strongly. Critically, previous work on APCS in TB (refs 52 and 53 in this paper) or other bacterial infections (p3 L23) has focussed on macrophages and macrophage deposition of ecm and so it seems strange to omit them from this work – mouse and human macrophage cell lines are readily available. Additionally previous work with PTX3 showed that this protein is produced and stored in macrophages as much as in neutrophils.

5 - Genetic risk of IPA in the transplant cohort p10. The p-values here are rather underwhelming given the clear and statistically well supported differences in the mouse model – presumably these differences could result from the mouse model being a complete homozygous gene KO whereas the human variants studied do not alter the protein sequence of the gene. Nevertheless a HR of 3.6 with p value of 0.012 for rs3753869 is strongly suggestive of some functional consequence. This needs to be clearly discussed in the paper. Is less protein or RNA produced by human cells/monocytes/PBMCs from individuals carrying either the heterozygous or homozygous SNP? This needs to be established. Presumably the authors have PBMCs from patients carrying the variants and it should be relatively straightforward to test this.

6 - Did the authors look for new rare SNPs in the gene that might be associated with infection by sequencing the APCS gene from individuals with IPA in their cohort? The SNPs tested are all common variants and there is a clear possibility that other undescribed variants with greater functional impact on gene function could exist in the cohort.

7 – Multivariate analysis – the authors have obviously performed this analysis given Table 2 and statements in the methods section but it is hard to find much mention in the text. Some further comment on this is needed – for example the significant variant rs2808661 is also associated with diabetes and polyneuropathic amyloidosis – it is difficult to determine whether such confounding effects were accounted for in the analysis – this needs a sentence or two of clarification in the text.

8 – Fig 2b. This is not clear. It is not really possible to distinguish spores in the image presented for review and there seems to be a large amount of non specific staining – two of the yellow arrows do not appear to point to anything? Spores are certainly visible in the inset...This either needs to be corrected with a higher quality image or omitted.

Reviewer #2 (*Aspergillus*, complement activation)(Remarks to the Author):

This is a highly original study reporting that the serum amyloid P component (SAP) is important in resistance against *Aspergillus fumigatus* both in humans and mice. It is well written and easy to understand.

I think the mice experiments look OK.

In the experiments addressing classical complement pathway activation using human serum, the authors do not discriminate between antibody-mediated activation and SAP mediated activation. I would assume that most human adults might have antibodies against aspergillus and since SAP circulates constitutively in the human situation, I think it would be interesting to see in human sera without anti-Aspergillus antibodies or antibodies as such and also sera depleted for SAP or blocking of the endogenous SAP how that will affect complement activation on *Aspergillus fumigatus*. I think it is important to clarify this in human plasma/sera, which might be very different from the mouse situation, which will clarify a role for SAP in an immunocompromised antibody situation.

I am not convinced about the genetic association study. We would have expected that the association would be with the recipient type because the site of SAP synthesis is the liver. No really functionality is provided with the SNPs are they associated with a difference in SAP protein levels? A more interesting approach could be to measure SAP plasma levels in the patients and see whether that might be associated with invasive pulmonary aspergillosis. I think this part of the study needs more elaboration.

Considering SAP as a therapeutic agent is fascinating, but again. The authors should reflect on that this is a normal plasma/serum protein in humans. Should this be administered in supraphysiological doses or maybe more interestingly to sites where the SAP concentration might low as in the lungs ???

Reviewer #3 (Anti-fungal immunity)(Remarks to the Author):

The article "Serum Amyloid P component is an essential element of resistance against *Aspergillus fumigatus*" by Doni et al represents an interesting study about the role of SAP in innate immune defence against invasive and inhalative aspergillosis. The authors show that SAP binds to fungal conidia, activates the complement cascade via classical pathway and increases phagocytosis of the spores. Comparing wild-type versus SAP-deficient mice the authors confirm the role of SAP in vivo. Findings that mutations in human SAP gene are associated with increased or decreased susceptibility complete this investigation.

The study is performed very clear and thorough. However, since data exist already for other pathogens, the role of SAP is not new and surprising. Therefore, this paper can be regarded to be beyond the scope of the journal and I recommend to reject the paper.

Minor criticism:

- infection dosage is extremely high and not very physiological
- study of different *Aspergillus* species does not include *A. terreus*, a frequent human pathogen
- purified SAP is applied intravenously to affect pulmonary infection; why not intratracheally?
- role of complement alternative pathway is not clarified; in addition, only MBL is investigated for lectin pathway, not other recognition molecules such as ficolins. For that reason also the role of lectin pathway remains unclear.
- in vitro experiments show SAP-mediated microbicidal activity of the spores. How can this be explained since there are not phagocytes?
- experiments in vitro are performed partly with murine, partly with human material (e.g. serum). It remains often unclear which material is from which species.

Re. Ms. # NCOMMS-19-38429-T entitled "Serum Amyloid P component is an essential element of resistance against *Aspergillus fumigatus*"

Point-by-point reply to Reviewers' comments:

Reviewer #1 (Anti-fungal immunity, genomics) (Remarks to the Author):

*The paper by Doni and colleagues describes the role of the pentraxin APCS in opsonisation - mediated defence against fungal infection. In many ways, these results follow previous work on PTX3 which was also shown to be an opsonisation protein for *Aspergillus* that led to disease susceptibility in KO mice – a phenotype that was also reversed by pre-opsonisation of spores with the protein (Nature. 420 (6912): 182–6). Here the authors demonstrate that APCS deficiency results in susceptibility to *Aspergillus fumigatus* infection in an immunocompromised mouse model and that resistance to infection can be restored by addition of exogenous APCS to the pathosystem. They also present evidence that variants in the gene are associated with invasive disease in humans.*

The work has been conducted by a world-class international team and is easily of a high enough standard for inclusion in Nature Communications. This is an important finding and has implications in understanding and treating fungal invasive disease. The paper is generally very well written, however there are a number of gaps in the paper that need to be addressed – in particular the paper contains no macrophage or other cell - type data, instead concentrating on neutrophils and there is no data on the function of the APCS variants in human cells. This data either needs to be included or its omission needs to be clearly justified before publication can be recommended.

We thank Reviewer #1 for his/her appreciation of our study (“*The work has been conducted by a world-class international team and is easily of a high enough standard for inclusion in Nature Communications. This is an important finding and has implications in understanding and treating fungal invasive disease.*”).

In his/her general comment Reviewer #1, and the Editor in the accompanying letter, raised the issue of a better characterization of immune cell function alteration and of the function of “APCS mutation” in mouse or human macrophages and DCs. On pages 10 and 11 and Figures 6 and 7 of the revised manuscript we address this important point. Briefly, we have used an *in vitro* and *ex vivo* assay to assess how SAP affects the function of mouse and human mononuclear phagocytes and DCs. SAP facilitated recognition of *A. fumigatus* conidia and phagocytosis by human monocytes and mouse macrophages and increased cytokine production. Similar results were obtained when preopsonized conidia were injected *in vivo* or when bone marrow-derived myelomonocytic cells were generated *in vitro* and tested using serum from APCS competent and incompetent mice. However, *in vivo* in lungs

of *Apcs*^{-/-} mice, we detected defective neutrophil phagocytosis whereas macrophages and DCs were unaffected. Thus, SAP has the capacity to facilitate phagocytosis by neutrophils and mononuclear phagocytes, but *in vivo* the major cell type affected by *Apcs* deficiency in lungs were neutrophils. No substantial impact of the *Apcs* on macrophage polarization was observed.

Specific comments to be addressed:

1 - The correct current gene name for SAP/PTX2 is APCS – this should be used throughout.

In agreement with Reviewer #1's comment, we removed "Pentraxin 2" or "PTX2" and replaced "SAP-deficient" with *Apcs*^{-/-} throughout the manuscript for simplification and accuracy in the gene nomenclature. We only mentioned PTX2 in the Abstract and Introduction because this acronym is used in some of the references, in particular in clinical trials.

2 - The section on human genetics needs considerable clarification and inclusion of data regarding numbers of individuals in the study that carried each variant, considerations of variant frequency in the general population and effect of the SNP on the protein. There are 483 individuals in the cohort or which 111 have IPA - how many carry the heterozygous/homozygous variants? This would allow an estimate of the importance of variants in this gene in disease. For Figure 5 the numbers of patients in each group are not clearly stated and need to be included in Figure 5 or figure legend – it certainly looks as if there are very few individuals in the homozygous variant groups which might account for the marginal p-values. A table in supplementary data would also be sufficient.

In his/her comment Reviewer #1 suggests to include data regarding numbers of individuals in the study that carried each variant and considerations of variant frequency in the general population to allow an estimate of the importance of variants in *Apcs* in disease. We thank Reviewer #1 for raising this important point. We have now included detailed information regarding the number of patients carrying each genotype among the IPA and control groups (new Table 2). In the new Fig. 9, the numbers indicate the total of patients with each genotype, regardless of whether they are cases of infection or controls, with the cumulative incidence providing an estimate of the number of affected individuals among the total of patients within each genotype. However, given Reviewer #1's suggestion, we have further detailed this information in the Figure Legend to new Fig. 9.

The prevalence of the variants in the study population needs to be discussed in the context of the consequence of carrying the SNP and in discussion of the prevalence of such variants in the general population – eg if the MAF for rs2808661 is 0.1 then homozygous variants will be rare in the population. Rs3753869 is more common and so on.

We have now added further discussion on this point (Discussion section, page 15). We are aware that the frequency of the mutant homozygous genotypes that are associated with the risk of IPA is relatively low. However, this fact further strengthens the critical role of these genetic variants as risk factors for IPA, since carriers of these genotypes reach cumulative incidence values of infection around 50%. This means that, despite being rare in the general population, they are markedly enriched among patients with IPA, thereby suggesting a relevant mechanism in the pathogenesis of the infection.

3 - It is not clear whether therapeutic potential can be realised easily. It is very encouraging that the protein is in human trials but some mention of how this would transfer to IPA is needed – how common do the authors really think this mechanism is in susceptibility to IPA? For example, allele frequencies for some SNPs in the population seem low and the mechanism might not be particularly common in IPA. Are the authors proposing a diagnostic or therapeutic avenue, or both? Is the proposal to screen donors for presence of the variant or to treat individuals already infected with injections of the protein? This could form an interesting and useful part of the discussion rather than the vague statements provided.

As stated above, the fact that rare homozygous genotypes of *APCS* variants are enriched in cases of infection supports its potential clinical applicability for screening donors, but most importantly, for managing the patients, e.g. for directing antifungal prophylaxis or intensifying fungal diagnostics in patients at higher genetic risk.

We also agree with Reviewer #1 on the need to try and define a diagnostic and therapeutic setting for SAP in IPA. Now this part is included in the Discussion section on page 15. The unexpected evidence of increased local levels of SAP in patients developing IPA, rather than in the circulation, suggests the development of a potential diagnostic tool to control the evolution of the disease. Pharmacodynamics studies conducted in patients affected by idiopathic fibrosis or in healthy donors

indicate a half-life in tissues of administered SAP considerably longer than 7 days and no serious adverse reactions at supra-physiological conditions (Refs. 34-37 quoted). In experimental IPA conducted in immunosuppressed mice, we observed therapeutic efficacy of SAP when administered systematically at same doses reported in mice (Refs. 32, 33 quoted) and in clinical settings in humans (Refs. 34-37 quoted). Therefore, we speculate about antifungal prophylaxis with SAP in patients undergoing hematopoietic stem-cell transplantation, with particular attention to those in which the variants associated to risk of infection is present (page 15).

4 - There is a strong focus on neutrophil responses in the pathosystem. I certainly agree that neutrophils are a major player in response to Aspergillus - and there are clear and credible results from the neutrophil work in this paper – but what about the other immune cells involved in the response? Is the effect of APCS opsonisation only on neutrophil phagocytosis or directly on neutrophil recruitment or does it for example stimulate DCs or Macrophages to emit neutrophil recruiting chemokines? The authors need to carefully consider this and justify their tight focus on neutrophils more strongly. Critically, previous work on APCS in TB (refs 52 and 53 in this paper) or other bacterial infections (p3 L23) has focused on macrophages and macrophage deposition of ecm and so it seems strange to omit them from this work – mouse and human macrophage cell lines are readily available. Additionally, previous work with PTX3 showed that this protein is produced and stored in macrophages as much as in neutrophils.

We concur with the opinion of Reviewer #1 and the Editor on the importance of an evaluation of an impact of SAP on functions of immune cells other than neutrophils. First, we conducted *ex vivo* and *in vivo* experiments of phagocytosis in different myeloid cell populations obtained from wt and *Apcs*^{-/-} mice (page 10-11 and new Fig. 6a and Fig. 7), as well as in purified human monocytes *in vitro* (new Fig. 6b). We then evaluated the immunomodulatory effects of SAP in bone marrow-derived macrophages in response to *A. fumigatus* conidia (new Fig. 6c). In the end, as indicated by Reviewer #1, in order to evaluate whether role of SAP in affecting alternative activation of macrophages was relevant in the context of a response to *A. fumigatus*, gene expression typically associated to M1 and M2 phenotypes was evaluated in wt and *Apcs*^{-/-} macrophages upon stimulation with fungus (not shown; discussed on page 10). Experiments were conducted in the presence of correspondent serum from wt and *Apcs*^{-/-} mice to preserve complement functions. The new results presented in the new Fig. 6 and Fig. 7, and discussed on pages 10 and 11, indicate that SAP affects the phagocytic capacity

of several immune cells, including monocytes and macrophages but not DCs, although neutrophils remain the major cell players in line with previous evidence supporting that neutrophils play a major role in resistance to *A. fumigatus* (new Ref. 49 quoted; York, *Nat. Rev. Microbiol.* 2018, PMID: 30177799; Mircescu, *et al. J. Infect. Dis.* 2009, PMID: 19591573).

5 - Genetic risk of IPA in the transplant cohort p10. The p-values here are rather underwhelming given the clear and statistically well supported differences in the mouse model – presumably these differences could result from the mouse model being a complete homozygous gene KO whereas the human variants studied do not alter the protein sequence of the gene. Nevertheless, a HR of 3.6 with p value of 0.012 for rs3753869 is strongly suggestive of some functional consequence. This needs to be clearly discussed in the paper. Is less protein or RNA produced by human cells/monocytes/PBMCs from individuals carrying either the heterozygous or homozygous SNP? This needs to be established. Presumably the authors have PBMCs from patients carrying the variants and it should be relatively straightforward to test this.

Reviewer #1 and Reviewer #2 raise the important issue that the association between genetic variants and disease incidence should reflect important functional consequences, although *APCS* rs2808661 and rs3753869 are synonymous SNPs. As shown in new Fig. 9c and discussed on pages 13 and 14, measurement of SAP in patients revealed that local (in BALFs) levels rather than systemic are correlated with IPA, and hence to represent a marker of disease. Donor *APCS* rs2808661 and rs3753869 SNPs are associated with serum levels of SAP (new Fig. 9d and Fig. 9e), suggesting a role in regulation of gene expression. In addition, prompted by the above observations we conducted a bioinformatics analysis in public databases and found that *APCS* is indeed expressed in murine and human immunocompetent cells (new Extended Data Fig. 12 discussed on page 14), including peripheral blood monocytes isolated from COVID-19 patients (new Extended Data Fig. 12f). In summary, the efforts conducted in response to the Reviewer's comments have shown that SAP/*APCS* is present in BALFs in IPA and that the genotype AA in donor *APCS* rs2808661 and rs3753869 SNPs is correlated with different levels of this molecule. Moreover, SAP can be induced in immunocompetent cells, including monocytes in COVID-19, and that these cells can represent a

relevant source of this defence molecule at local tissue sites, complementing systemic non-hematopoietic cell-derived protein.

6 - Did the authors look for new rare SNPs in the gene that might be associated with infection by sequencing the APCS gene from individuals with IPA in their cohort? The SNPs tested are all common variants and there is a clear possibility that other undescribed variants with greater functional impact on gene function could exist in the cohort.

We have not performed a screening for the presence of rare variants in *APCS*. We have instead decided to follow a strategy based on the analysis of common variation spanning the entire gene using a haplotype-based tagging approach. This was performed since we are aware that our current sample size is modest to detect accurate associations of very rare variants. This however does not preclude that other, rare variants, may also exist with strong impact on *APCS* function and that may be important for the pathogenesis of IPA.

7 – Multivariate analysis – the authors have obviously performed this analysis given Table 2 and statements in the methods section but it is hard to find much mention in the text. Some further comment on this is needed – for example the significant variant rs2808661 is also associated with diabetes and polyneuropathic amyloidosis – it is difficult to determine whether such confounding effects were accounted for in the analysis – this needs a sentence or two of clarification in the text.

We agree with the Reviewer's suggestion that this is an important point to consider. We have included all the clinical factors independently associated or tending ($P < 0.15$) toward risk of IPA in the multivariate analysis (Table 3). Of note, we had no information on the presence of diabetes or polyneuropathic amyloidosis since these are not considered "classical" risk factors for IPA in the stem-cell transplantation setting. Although genetic variation in *APCS* has been evaluated in the context of other diseases, including type 2 diabetes and polyneuropathic amyloidosis (new Ref. 59 quoted), these are not acknowledged as relevant conditions predisposing to IPA. This point is now discussed on page 15.

8 – Fig 2b. This is not clear. It is not really possible to distinguish spores in the image presented for review and there seems to be a large amount of non-specific staining – two of the yellow arrows do

not appear to point to anything? Spores are certainly visible in the inset...This either needs to be corrected with a higher quality image or omitted.

We agree with the concerns raised by Reviewer #1 regarding clarity of the image proposed in the previous version of the manuscript. Now, we show images obtained from a new experiment in which FITC-labelled *A. fumigatus* conidia were injected into the lungs of mice and samples for confocal microscopy were prepared and stained as described in the Materials and Methods section (new Fig. 2b, Microscopy section in Materials and Methods page 35). Prompted by Reviewer #1's comment, this new strategy provided better evidence of colocalization between SAP and conidia is better evident. A lung of an *Apcs*^{-/-} mouse is now shown as control of SAP specific staining. Images related to other controls, such as isotypic IgGs, is available as raw data in the Source Data File in the worksheet Fig. 2.

Reviewer #2 (Aspergillus, complement activation) (Remarks to the Author):

This is a highly original study reporting that the serum amyloid P component (SAP) is important in resistance against Aspergillus fumigatus both in humans and mice. It is well written and easy to understand.

I think the mice experiments look OK.

In the experiments addressing classical complement pathway activation using human serum, the authors do not discriminate between antibody-mediated activation and SAP mediated activation. I would assume that most human adults might have antibodies against aspergillus and since SAP circulates constitutively in the human situation, I think it would be interesting to see in human sera without anti-Aspergillus antibodies or antibodies as such and also sera depleted for SAP or blocking of the endogenous SAP how that will affect complement activation on Aspergillus fumigatus. I think it is important to clarify this in human plasma/sera, which might be very different from the mouse situation, which will clarify a role for SAP in an immunocompromised antibody situation.

We thank Reviewer #2 for his/her appreciation of our work (“*This is a highly original study reporting that the serum amyloid P component (SAP) is important in resistance against Aspergillus fumigatus both in humans and mice. It is well written and easy to understand. I think the mice experiments look OK*”).

We thank Reviewer #2 for raising the important issue of discriminating between antibody-related and SAP-mediated functions in the response to *A. fumigatus*. As discussed on pages 8 and 9-10 and shown in the new Figure 4, experiments using plasma from healthy donors depleted of selected components showed that both endogenous IgGs and SAP are important for neutrophil phagocytosis (new Fig. 4b) and complement deposition on *A. fumigatus* conidia (new Fig. 4c). As shown now in Fig. 3d and discussed on page 8, SAP-mediated opsonophagocytosis and complement activation did not involve an antibody-mediated engagement of Fc γ R_s. Accordingly, SAP exhibited a potential therapeutic effect in experimental aspergillosis models in mice under cyclophosphamide-induced immunosuppression, associated to a reduction in circulating levels of IgGs (not shown, available in the Source Data File at worksheet Fig. 8).

I am not convinced about the genetic association study. We would have expected that the association would be with the recipient type because the site of SAP synthesis is the liver. No really functionality is provided with the SNPs are they associated with a difference in SAP protein levels? A more interesting approach could be to measure SAP plasma levels in the patients and see whether that might be associated with invasive pulmonary aspergillosis. I think this part of the study needs more elaboration.

Reviewer #1 and Reviewer #2 raise the important issue that the association between genetic variants and disease incidence should reflect important functional consequences, although *APCS* rs2808661 and rs3753869 are synonymous SNPs. As shown in new Fig. 9c and discussed on pages 13 and 14, measurement of SAP in patients revealed that local (in BALFs) levels rather than systemic are correlated with IPA, and hence to represent a marker of disease. Donor *APCS* rs2808661 and rs3753869 SNPs are associated with the lower serum levels of SAP (new Fig. 9d and Fig. 9e), suggesting a role in regulation of gene expression. In addition, prompted by the above observations we conducted a bioinformatics analysis in public databases and found that *APCS* is indeed expressed in murine and human immunocompetent cells (new Supplementary Fig. 12 discussed on page 14), including peripheral blood monocytes isolated from COVID-19 patients (new Extended Data Fig. 12f). In summary, the efforts conducted in response to the reviewer's comments have shown that

SAP/APCS is present in BALFs in IPA and that the genotype AA in donor APCS rs2808661 and rs3753869 SNPs is correlated with different levels of this molecule. Moreover, SAP can be induced in immunocompetent cells, including monocytes in COVID-19, and that these cells can represent a relevant source of this defence molecule at local tissue sites, complementing systemic non-hematopoietic cell-derived protein.

Considering SAP as a therapeutic agent is fascinating, but again. The authors should reflect on that this is a normal plasma/serum protein in humans. Should this be administered in supraphysiological doses or maybe more interestingly to sites where the SAP concentration might low as in the lungs???

Reviewer #2 raises that important issue of local versus systemic administration of SAP in a therapeutic perspective. This point is now addressed in new Fig. 8c and Fig. 8d and on page 12. Systemic administration of supraphysiological doses of SAP was based on previous preclinical and clinical therapeutic trials on lung fibrosis (Refs. 32-37 quoted). Prompted by the Reviewer's comments we performed a therapeutic experiment with intratracheal administration of SAP (new Fig. 8c). The results obtained prove the principle that local administration of SAP has indeed therapeutic potential.

Reviewer #3 (Anti-fungal immunity) (Remarks to the Author):

The article "Serum Amyloid P component is an essential element of resistance against Aspergillus fumigatus" by Doni et al represents an interesting study about the role of SAP in innate immune defence against invasive and inhalative aspergillosis. The authors show that SAP binds to fungal conidia, activates the complement cascade via classical pathway and increases phagocytosis of the spores. Comparing wild-type versus SAP-deficient mice the authors confirm the role of SAP in vivo. Findings that mutations in human SAP gene are associated with increased or decreased susceptibility complete this investigation.

The study is performed very clear and thorough. However, since data exist already for other pathogens, the role of SAP is not new and surprising. Therefore, this paper can be regarded to be beyond the scope of the journal and I recommend to reject the paper.

We thank Reviewer #3 for his/her appreciation of our work ("The study is performed very clear and thorough"). The Reviewer #3 raises a general issue as to the novelty of the paper, given the fact that "data exist already for other pathogens". The Reviewer is right and, in the paper, we quote previous

work on other pathogens (Refs. 15-18, 25 and 26 quoted). In spite of this duly quoted previous knowledge, we feel that the paper conveys substantial new knowledge which can be summarized as follows:

- Role of SAP in fungal infections for which there is no precedent;
- Underlying mechanisms;
- Genetic association in human disease, for which there is no precedent;
- Therapy in a relevant preclinical model, with SAP being in clinical trials for lung fibrosis with positive results.

Minor criticism:

- *infection dosage is extremely high and not very physiological*

The infection dose used in the present study is similar to the ones used in previous studies by us and by others (Ref. 38 quoted, Garlanda, *et al. Nature* 2002, PMID: 12432394; Jaillon, *et al. J. Exp. Med.* 2007, PMID: 17389238; Bozza, *et al. J. Immunol.* 2008, PMID: 18322211) including a comparative medicine context (Desoubeaux, *et al. Comp. Med.* 2018, PMID: 29663936). A high dose if anything raises the bar for therapeutic efficacy.

- *study of different Aspergillus species does not include A. terreus, a frequent human pathogen.*

Prompted by the Reviewer's comment, we have expanded the studies on the role of SAP in response to different clinically relevant species of the *Trichocomaceae* family, such as *A. flavus* (new Extended Data Fig. 1 and Fig. 6) and *A. terreus* (new Extended Data Fig. 2 and Fig. 6), in addition to having added results omitted in the previous version of the manuscript on the non-relevance of SAP in a *Candida albicans* dissemination model (new Extended Data Fig. 3 and Fig. 6).

- purified SAP is applied intravenously to affect pulmonary infection; why not intratracheally?

We agree with Reviewer #3 on the importance of testing the therapeutic efficacy of locally administered SAP in the experimental model of IPA, a condition that could mimic clinical treatment in patients through aerosol. In the new Figure 8c and 8d, we show one of two experiments (the raw data of the second shown in the Source Data File at worksheet Fig. 8) performed in mice after immunosuppressive treatment and intratracheal administration of SAP. The results obtained suggest that SAP has therapeutic activity, when given intratracheally, similar to that by the i.v. route.

- role of complement alternative pathway is not clarified; in addition, only MBL is investigated for lectin pathway, not other recognition molecules such as ficolins. For that reason, also the role of lectin pathway remains unclear.

Reviewer #3 raises the important issue of the role of the lectin and alternative pathway of complement activation. To address this important issue, in addition to previous results on neutrophil phagocytosis (previous Fig. 2i and Fig. 2l, new Fig. 3a and Fig. 3b) and complement deposition (previous Fig. 3, new Fig. 5) performed with human sera depleted of different complement molecules, we now show new experiments of neutrophil phagocytosis in blood of mice defective for the lectin (*Mbl1/2^{-/-}*) or alternative (*Fb^{-/-}*) complement pathway (new Fig. 3b). These new results, in addition to the previous ones of the abolition of SAP-mediated responses in human serum in the absence of C1q and in line with reports in literature on other pathogens (Ref. 16 quoted), clearly indicate that the interaction with the classical complement pathway is required for the initiation of SAP-mediated phagocytosis and complement activation.

In previous studies, SAP did not interact with ficolins to form functional heterocomplexes important to signal sequestration of altered self-cells and *A. fumigatus*, as other pentraxins (Ma, *et al.*, *J. Immunol.* 2013, PMID: 23817411). In an effort to understand a possible relevance of ficolins in SAP-mediated phagocytosis, isolated human neutrophils were challenge with *A. fumigatus* conidia pre-opsonized or not with human SAP in the presence of autologous serum and of mAbs anti-M- or L-

ficolin. As shown below, incubation with anti-ficolin mAbs did not modulate phagocytosis by neutrophils from two donors, thus suggesting no relevance. This result is briefly mentioned on page 8 and new new Extended Data Fig. 8.

***In vitro* phagocytosis of *A. fumigatus* by human neutrophils in the presence of anti-M- or anti-L-ficolin mAbs.** FACS analysis of phagocytosis (30 min) by freshly isolated human neutrophils (1×10^5). FITC-labelled AF conidia (1×10^6) were pre-opsonized or not with human SAP (2×10^8 conidia and $100 \mu\text{g}$ human native SAP). Phagocytosis was performed in the presence of 10% autologous serum and of mAbs ($10 \mu\text{g}/\text{ml}$) anti-M-(#036-051 1, mouse IgG₁) and L-(-#FCN219, mouse IgG_{2A}) ficolin (all from Santa Cruz Biotechnology) or an irrelevant mouse IgG₁ (ThermoFisher Scientific). Each spot corresponds to a replicate of an experiment performed in quadruplicate.

- *in vitro* experiments show SAP-mediated microbicidal activity of the spores. How can this be explained since there are not phagocytes?

Reviewer #3 raises the issue of phagocytosis-independent microbicidal function of SAP. Extracellular killing via complement activation and MAC formation has previously been reported for fungi including *A. fumigatus* (Bidula *et al.*, *J. Infect. Dis.* 2015, PMID: 25612732; Kozel, *Clin.*

Microbiol. Rev. 1996, PMID: 8665475). We report microbicidal activity in a classical cell viability assay (previous Fig. 3d, new Fig. 5d) and CFUs counts (previous new Extended Data Fig. 6, new new Extended Data Fig. 9) in conditions where SAP opsonisation increased MAC deposition on *A. fumigatus* (previous Fig. 3c, new Fig. 5c). Posaconazole was used as control in the experiments with the resazurin-based cell viability assay and in CFU counts (new Fig. 5d and new Extended Data Fig. 9). An analysis of what appears to be a minor pathway of resistance was beyond the scope of the paper.

- experiments in vitro are performed partly with murine, partly with human material (e.g. serum). It remains often unclear which material is from which species.

As suggested by Reviewer #3, we have now specified the human or mouse material used throughout the manuscript.

Reviewers' comments:

Reviewer #2 (Remarks to the Author):

I am satisfied with the reply of authors.

I have read through the rebuttal from the authors and I think they have done a thorough work also in answering referee 1.